



# The nano-scanning electrical mobility spectrometer (nSEMS) and its application to size distribution measurements of 1.5-25 nm particles

Weimeng Kong[1,a], Stavros Amanatidis[1], Huajun Mai[2], Changhyuk Kim[1,b], Benjamin C. Schulze[2], Yuanlong Huang[2], Gregory S. Lewis[3], Susanne V. Hering[3], John H. Seinfeld[1,2], and Richard C. Flagan[1,2]

[1]Division of Chemistry and Chemical Engineering, California Institute of Technology, Pasadena, CA 91125
[2]Department of Environmental Science and Engineering, California Institute of Technology, Pasadena, CA 91125
[3]Aerosol Dynamics Inc., Berkeley, CA
[a]now at: California Air Resources Board, Sacramento, CA 95814
[b]now at: Department of Environmental Engineering, Pusan National University, Busan, Republic of Korea
**Correspondence:** Richard. C. Flagan (flagan@caltech.edu)

**Abstract.** Particle size measurement in the low nanometer regime is of great importance to the study of cloud condensation nuclei formation and to better understand aerosol-cloud interaction. Here we present the design, modeling, and experimental characterization of the nano-scanning electrical mobility spectrometer (nSEMS), a recently developed instrument that probes particle physical properties in the 1.5 - 25 nm range. The nSEMS consists of a charge conditioner, a novel differential mobility analyzer, and a two-stage condensation particle counter (CPC). The charge conditioner employs a soft x-ray bipolar ion source in a compact housing designed to optimize both nanoparticle charging and transmission efficiency. The mobility analyzer, a radial opposed migration ion and aerosol classifier (ROMIAC), can classify nanometer-sized particles with minimal degradation of its resolution or diffusional losses. The ROMIAC operates on a dual high-voltage supply with fast polarity-switching capability to minimize sensitivity to variations in the chemical nature of the ions used to charge the aerosol. Particles transmitted through the charge conditioner and mobility analyzer are measured using a two-stage CPC. They are first activated in a fast-mixing diethylene glycol (DEG) stage before being counted by a second detection stage, an ADI MAGIC[TM] water-based CPC. The transfer function of the integrated instrument is derived from both finite-element modeling and experimental characterization. The nSEMS performance has been evaluated during measurement of transient nucleation and growth events in the CLOUD atmospheric chamber at CERN. We show that the nSEMS can provide high time and size resolution measurement of nanoparticles, and can capture the critical aerosol dynamics of newly formed atmospheric particles.

## 1 Introduction

Aerosol particles can either be emitted into the atmosphere directly from primary sources, or generated through the nucleation of atmospheric condensable precursor vapors. Atmospheric nucleation, or new particle formation (NPF), is frequently observed across the globe under diverse environmental conditions, ranging from populated urban centers (Dunn et al., 2004; Wang et al., 2017; Yao et al., 2018; Wang et al., 2020) to remote areas such as forests or oceans (O'Dowd et al., 2002; Bonn and Moortgat, 2003; Paasonen et al., 2010; Dall'Osto et al., 2017), as well as in the free troposphere (Kulmala et al., 2004; Kerminen et al.,





2018; Lee et al., 2019). Large-scale simulations and parameterizations suggest that NPF may generate half of the global cloud condensation nuclei (CCN, Merikanto et al. (2009); Gordon et al. (2017)). In addition to their climate impact, ultrafine particles formed by nucleation can also have disproportionate adverse effects on human health (Brown et al., 2000).

To understand the formation and subsequent growth of freshly nucleated particles in the atmosphere, and to evaluate their impact on climate and human health requires measurement techniques that can enable particle size distribution measurements in the low-nanometer regime. Nanoparticle sizing is often achieved using electrical mobility to separate charged particles according to the velocities with which they migrate in an electric field. The differential mobility analyzer (DMA) has long been the most widely used instrument to measure size distribution of sub-micrometer ($< 1 \ \mu$m) aerosol particles (Knutson and

Whitby, 1975; Flagan, 1998). Most DMAs separate charged aerosol particles of different electrical mobilities by applying an electric field between two coaxial cylindrical electrodes that are parallel to a particle-free sheath flow. Classified particles from the DMAs are typically counted by a condensation particle counter (CPC) that uses supersaturated vapors to grow particle seeds before detection (Quant et al., 1992). This integrated instrument initially used step-wise variation in the voltage. By keeping the voltage applied between the electrodes constant at each step, particles are transmitted throughout the entire system under

a steady field profile before they are counted by the CPC. This static-mode measurement system is referred to as the DMPS, or the differential mobility particle sizer (Fissan et al., 1983). However, since the DMPS requires the system to reach a steady-state before a reliable measurement can be made, the relatively long residence time of the particles in the DMA makes it less ideal capturing transient aerosol activities. By continuously changing the voltage through an exponential ramp, the scanning electrical mobility spectrometer (SEMS; also known as the scanning mobility particle sizer, SMPS) considerably accelerates

the particle classification using electrical mobility (Wang and Flagan, 1990).

While the traditional DMPS, SMPS, or SEMS systems can only detect particles with mobility-equivalent diameters above 10 nm, many efforts have focused on extending the classification range to smaller aerosol particles. The first major step in this direction was the Vienna short-column DMA that measured nanoparticles down to 3.5 nm (Winklmayr et al., 1991); the "nano-DMA", modified from the traditional long-column DMA design, can measure particle size distribution in the range of 3

- 50 nm (Chen et al., 1998); a radial DMA (RDMA) that classifies particles in a radial flow toward the center of parallel disk electrodes also demonstrated a high detection efficiency for particles between 3 - 10 nm (Zhang et al., 1995). Both designs have been extended to smaller sizes. Through meticulous aerodynamic design and very large sheath flow rates, up to 1000 L/min, Rosell-Llompart et al. (1996) refined the Vienna DMA into one instrument suitable for particle classification down to as small as 1 nm. Brunelli et al. (2009) developed an RDMA that could also classify 1 nm particles at much more modest flow

rates, albeit at lower resolution. An alternate mobility analyzer design, the opposed migration aerosol classifier (OMAC), uses a particle-free cross flow instead of the parallel sheath flow in the DMAs to balance particle electrical migration. Since the aerosol being classified fills the space between the electrodes, rather than occupying only a narrow slice of that space as in a DMA, this changes of the scaling for where diffusion begins to degrade the resolution of the classifier, thereby enabling classification at lower voltages, or operation at higher resolution than is possible with conventional DMAs (Flagan, 2004; Downard et al.,

2011). A radial-flow form of this instrument, the radial opposed migration ion and aerosol classifier (ROMIAC), has proven capable of measuring sub-2 nm particles or ions, and even separating peptide stereoisomers owing to its high resolving power





(Mui et al., 2013, 2017). The challenges with measuring particles in the low-nanometer regime lie not only in classification, but also in particle detection. Some single-stage CPCs have been operated at sufficiently high supersaturation to activate particles as small as 1 nm diameters, but in the experiment for which this instrument has been developed, where measurements must

be made in a high-radiation environment, this can lead to nucleation within the CPC. Therefore, we took a more conservative approach that has proven robust and effective for sub-10nm particle detection, namely a two-stage CPC, in which the first stage employs a low vapor pressure working fluid, typically diethylene glycol (DEG) that can activate small particles with minimal risk of homogeneous nucleation (Iida et al., 2009). Owing to the low vapor pressure, the first stage does not grow particles to optically detectable size, so a second "booster" stage is used to grow the activated clusters and optically detect them. The

second stage is typically a conventional CPC. The operation of two activation and growth systems in series compounds another challenge to SEMS/SMPS measurements; the residence time within the CPC can distribute counts of particles that exit the DMA over many time bins (Russell et al., 1995; Collins et al., 2002), thereby degrading the resolution of the instrument. This effect becomes increasingly important at scan rates that are fast relative to the response time of the CPC. Therefore, CPCs with a narrow distribution of residence times are preferred particularly for one SEMS that targets the low-nanometer range, in which

the resolution is also compromised by Brownian diffusion.

In this work we show the development of a nano-scanning electrical mobility spectrometer (nSEMS) that features a soft x-ray aerosol charge conditioner, a fast-scanning ROMIAC particle classifier, and a two-stage CPC, to acquire fast and accurate particle size distributions in the range of 1.5 - 25 nm. The two-stage CPC includes a fast-mixing activation stage using DEG as working fluid, followed by an eco-friendly, fast-response, water-based CPC (Hering et al., 2019). Each component of the

nSEMS was characterized separately; the integrated transfer function was derived based on both experimental results and finite-element modeling using COMSOL Multiphysics[TM]. The nSEMS has been intensively used in the Cosmics Leaving OUtdoor Droplets (CLOUD) experiments at CERN, in which its size resolution and fast response have made it possible to follow very rapid growth of freshly nucleated nanoparticles, and to identify a new mechanism for new particle formation in highly polluted atmosphere (Wang et al., 2020). A comparison of nSEMS data with measurements from other well-calibrated particle sizing

instruments at CLOUD confirms its capacity to provide reliable size distribution in the low-nanometer size regime.

## 2   The nSEMS design and system features

The nSEMS was designed to capture critical aerosol dynamics during atmospheric nucleation and subsequent nanoparticle growth, both in environmental chamber experiments and in ambient measurements. To this end, its design and operating parameters have been optimized to provide size distribution measurements with relatively high size resolution in the sub-25

nm range, and with a fairly short duty cycle. The nSEMS classifies particles of different sizes according to their electrical mobilities, $Z_p$, which is defined as the ratio of particle migration velocity, $v_m$, to the electric field strength within the classifier, $E$:

$$Z_p = \frac{v_m}{E} = \frac{\phi e C_c}{3\pi\mu d_p} \tag{1}$$



**Table 1.** Default nSEMS operating parameters, optimized for measurements of NPF events and nanoparticle growth. These settings enable particle size distribution measurements in the range of 1.5 - 25 nm, with a duty cycle of 1 min and a size classification resolution of $\mathcal{R}_{\mathrm{nom,nd}}$ = 10.

| Parameter | Notation | Value |
|---|---|---|
| Instrument total sampling rate (L/min) | $Q_\mathrm{s}$ | $4.60^a$ |
| ROMIAC polydisperse flow rate (L/min) | $Q_\mathrm{a}$ | 1.20 |
| ROMIAC monodisperse flow rate (L/min) | $Q_\mathrm{c}$ | 1.20 |
| ROMIAC incoming cross-flow flow rate (L/min) | $Q_\mathrm{x, in}$ | 12.0 |
| ROMIAC outgoing cross-flow flow rate (L/min) | $Q_\mathrm{x, out}$ | 12.0 |
| DEG feeding rate (L/min) | $Q_\mathrm{DEG}$ | 0.30 |
| CPC sampling rate (L/min) | $Q_\mathrm{CPC}$ | $1.00^b$ |
| Low electrode voltage (V) | $V_\mathrm{low}$ | 20.0 |
| High electrode voltage (V) | $V_\mathrm{high}$ | 10,000 |
| CPC sample conditioner temperature (°C) | $T_\mathrm{co}$ | $20^c$ |
| CPC DEG saturator temperature (°C) | $T_\mathrm{sat}$ | 70 |
| CPC sample condenser temperature (°C) | $T_\mathrm{cond}$ | 10 |
| Voltage ramp time (s) | $t_\mathrm{ramp}$ | 50 |
| Holding time at $V_\mathrm{low}$ (s) | $t_\mathrm{low}$ | 4 |
| Holding time at $V_\mathrm{high}$ (s) | $t_\mathrm{high}$ | 2 |
| Scan duty cycle (s) | $t_\mathrm{total}$ | 60 |
| CPC data recording frequency (s) | $t_\mathrm{c}$ | 0.20 |

$^a$ All flow rate measurements have an uncertainty of $\pm 2\%$.

$^b$ The MAGIC$^{\mathrm{TM}}$ water-based CPC is a special, high-flow rate concise CPC.

$^c$ All temperature measurements have an uncertainty of $\pm 0.1°$C.

where $\phi$ is the net number of elementary charges, $e$, on the particle, $C_c$ is the Cunningham slip correction factor that accounts for the noncontinuum effects, $\mu$ is the dynamic viscosity of air, and $d_p$ is the particle diameter. Figure 1 shows a schematic of nSEMS main components; a soft x-ray charger, a ROMIAC that operates on a continuously varying voltage, and a two-stage CPC. Detailed operating parameters and default settings are summarized in Table 1. Particles entering a mobility-based sizing system typically pass through a charge conditioner to attain a known, steady-state charge distribution to enable accurate data inversion. The nSEMS charge conditioner employs a soft x-ray source that directly ionizes the air around the incoming particles. The charge conditioner can be remotely switched off to enable measurements based on the natural charge of ions or particles, if so desired. Although it remains challenging to determine the actual charging state of sub-20 nm particles (Lòpez-Yglesias and Flagan, 2013), the Fuchs charging efficiencies, $f_c$, used in electrical mobility calculations are approximated based on the Wiedensohler (1988) correlation for consistency with other studies (Tröstl et al., 2015; Stolzenburg et al., 2017) and to facilitate comparison with data from other instruments. While the ROMIAC only requires an aerosol sample flow rate of





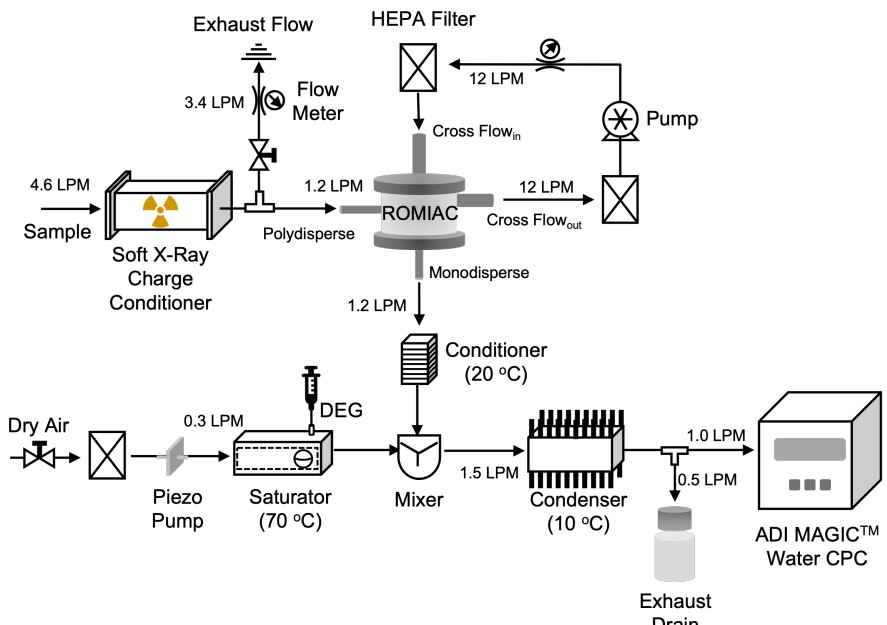

**Figure 1.** Schematic of the nSEMS main components; a soft x-ray charge conditioner, a radial opposed-migration ion and aerosol classifier (ROMIAC), and a two-stage condensation particle counter (CPC). Core-sampling of the input sample flow is employed to minimize particle diffusional losses in the charge conditioner. The ROMIAC is operated on exponentially increasing voltage ramps between 20 V and 10,000 V of both positive and negative polarity, at 1.2 L/min and 12 L/min aerosol and cross-flows (Table 1). After exiting the classifier, monodisperse aerosol particles are mixed with 0.3 L/min diethylene glycol (DEG) vapor in the first stage of the CPC. This fast-mixing stage allows nanoparticles to grow through rapid vapor condensation before they enter the second, "booster" CPC stage, a modified Aerosol Dynamics Inc. (ADI) MAGIC™ water CPC operated at a flow rate of 1.0 L/min, where particles grow further and are optically detected.

1.2 L/min in the default setup, the charge conditioner pulls the aerosol at a higher flow rate (4.6 L/min) to overcome the high diffusional losses. The aerosol entering the ROMIAC is drawn from the core of the flow exiting the charge conditioner (Stolzenburg and McMurry, 1991). The charged nanoparticles transmitted through a core-sampling probe enter the ROMIAC at a flowrate of 1.2 L/min, tangentially into a flow distribution channel, and then enter the classification region through a narrow slit. Unlike traditional DMAs, in which the particle-free sheath flow is parallel to the electrodes, the ROMIAC uses
a 12 L/min recirculated, filtered cross-flow that enters and exits the classification region through screen electrodes (Flagan, 2004; Mui et al., 2013, 2017). For nondiffusive particles, the ideal resolving power of the classifier, $\mathcal{R}_{nom,nd}$, is given as:

$$\mathcal{R}_{nom,nd} = \frac{1}{\beta\left(1 + |\delta|\right)} \tag{2}$$

for all of the designs, configurations or flow rate ratios (Flagan, 1999). The two flow factors, the imbalance factor, $\delta$, and the aerosol-to-cross-flow ratio, $\beta$, are defined as:

$$\delta = \frac{Q_c - Q_a}{Q_a + Q_c}, \; \beta = \frac{Q_a + Q_c}{Q_{x,\,in} + Q_{x,\,out}} \tag{3}$$





where $Q_a$ is the incoming polydisperse aerosol flow rate, $Q_c$ is that of the outgoing classified sample flow, and $Q_{x,\,in}$ and $Q_{x,\,out}$ are the entering and exiting cross flow rates. If the incoming flows are balanced with the outgoing flows ($Q_a = Q_c$; $Q_{x,\,in} = Q_{x,\,out}$), then the two factors can be simplified to $\delta = 0$, $\beta = \mathcal{R}_{nom,nd}^{-1}$. A resolution of $\mathcal{R}_{nom,nd} = 10$ is generally sufficient to capture the critical cluster-to-ion formation process under ambient conditions. In order to get a reasonable size coverage to study the subsequent condensational or coagulational growth of newly formed nanoparticles, voltage is exponentially ramped between 20 V and 10,000 V with a characteristic time:

$$\tau_s = \frac{t_{ramp}}{\ln(V_{high}/V_{low})} \tag{4}$$

of approximately 8 s. The mean residence time of the sample flow in the classifier, $\tau_f$, is:

$$\tau_f = \frac{\mathcal{V}_{class}}{Q_{class}} \tag{5}$$

with $\mathcal{V}_{class}$ and $Q_{class} = (Q_a + Q_c)/2$ corresponding to the volume and aerosol flow rates in the classification region. For the ROMIAC, $\mathcal{V}_{class} = \pi(R_2^2 - R_1^2)b$, where $R_1 = 0.24$ cm and $R_2 = 1.61$ cm are the inner/outer electrode radii, $b = 1$ cm is the gap between the high voltage and ground electrodes, resulting in a classification volume $\mathcal{V}_{class} \simeq 8.0$ cm$^3$. The resulting mean gas residence time at the nominal aerosol flow rate is $\tau_f \simeq 0.4$ s. The mobility of the particles that are transmitted through an ideal, constant-voltage ROMIAC is:

$$Z_{p,ideal}^* = \frac{(Q_{x,\,in} + Q_{x,\,out})\,b}{2\pi\,(R_2^2 - R_1^2)\,V^*} \tag{6}$$

where $Z_{p,ideal}^*$ can be treated as the centroid particle electrical mobility in scanning mode assuming highly idealized flow and electric fields and corresponds to the peak of the transfer functions (Zhang et al., 1995; Mui et al., 2017). $V^*$ is the corresponding voltage applied to the central electrode when particles are detected by the CPC.

Most mobility-based particle sizing systems measure only one polarity of charged particles (usually positive) by employing a single polarity high voltage supply. Since the nSEMS employs a soft x-ray ion source in the charge conditioner, the aerosol downstream of the charge conditioner contains both negatively and positively charged particles as they collide with ions of both polarities. Because ion properties, such as mass, mobility, and concentration, as well as experimental conditions can all affect particle charging efficiency, and the ions produced by the soft x-rays can vary due to trace species in the gas, measuring only particles with single polarity may lead to uncertainties and variabilities in computing particle concentrations (Steiner et al., 2014; Chen et al., 2018). To optimize instrument performance and avoid potential variability in particle charging, the ROMIAC operates on a custom-built dual high-voltage supply with fast polarity-switching capability. In the default operating mode of the nSEMS, the polarity of the scanning voltage is switched at the start of every scan, but the polarity can also be fixed, either positive or negative, or it can be turned off to meet different scientific needs. This feature not only helps to better understand the performance of bipolar diffusion charging, it also enables measurement of the charge state of the sampled aerosol particles by deactivating the charge conditioner for some or all scans. This becomes an important feature when studying atmospheric nucleation as it enables discrimination between neutral and ion-mediated nucleation (Kirkby et al., 2016; Wagner et al., 2017).

Classified particles transmitted through the ROMIAC are subsequently detected by a two-stage CPC that enables particle counting approaching 1 nm in size (Iida et al., 2009). The first stage employs a fast-mixing condensational activation and





growth reactor (Wang et al., 2002; Shah and Cocker, 2005) that uses DEG as the working fluid to activate the nanoparticles.

As in the Airmodus particle-size magnifiers and the CPC of Sgro and Mora (2004), supersaturation is produced by turbulent mixing of the hot (70 °C) DEG vapor with cold (20 °C) particle-laden flow, at a flow rate of 0.3 and 1.2 L/min, respectively. The downstream growth tube is cooled to 10 °C to accelerate particle growth and remove excess vapor. In contrast to the Airmodus particle size magnifier (PSM) and the CPC of Sgro and Mora (2004), on which the PSM is based, the residence time in the activation stage of the present CPC has been minimized to speed instrument response. A modified ADI MAGIC$^{TM}$ water-

based CPC serves as the second stage to grow particles sufficiently large for optical detection (Hering et al., 2019). Particle counts are recorded over the nSEMS size distribution scan at 5Hz. The sample flow rate of the CPC is 1.00 L/min. Between the activation and booster stages, the flow is split between the water-CPC and a smaller excess flow to minimize deposition of excessive DEG vapor in the intervening plumbing and the water CPC.

Data acquisition and instrument control for the nSEMS are accomplished with a National Instruments$^{TM}$ sbRIO-9637 Com-

pactRIO single-board controller coupled with a Field Programmable Gate Array (FPGA) module. The FPGA module, which is programmable using LabVIEW 2018, is capable of operating at clock speeds up to 40 MHz with optimized hardware and memory settings. The FPGA is controlled by a microprocessor that runs on a real-time Linux operating system (OS), greatly reducing the overhead and response lag associated with typical LabVIEW applications running on other platforms. The real-time OS and FPGA enable independent time loops for precise control of the voltage exponential ramp and recording of CPC concen-

trations. The board controller is connected to a Windows PC via Ethernet, enabling communication among different programs, visualization of real-time data, and online monitoring of critical parameters without compromising instrument timing.

## 3   Characterization of the nSEMS

Compared to the DMPS, the SEMS can accelerate mobility-based size distribution measurements by classifying particles in a time-varying electric field, and eliminating the transition time between measurement channels. Although the exponential

voltage ramping allows investigation of rapidly evolving aerosol particles, it alters the particle trajectories in the classifier, such that the transfer function may differ significantly from that expected for a DMPS. Numerical simulations of particle trajectories in a scanning cylindrical DMA have shown that the width of instrument transfer function for fast scans can be significantly greater than that for a static (constant voltage) DMA (Collins et al., 2004; Mai and Flagan, 2018; Mai et al., 2018). Similarly, voltage scanning of the ROMIAC may distort the transfer functions from those seen in static-mode operation.

### 3.1   Finite-element modeling of particle transmission

Both numerical simulations and derivations of analytical solutions for idealized instruments have proven to be powerful tools in the study of the transfer functions of DMAs operating in scanning mode (Collins et al., 2004; Dubey and Dhaniyala, 2011). However, the ROMIAC geometry and particle trajectories are more complicated than those in long-column cylindrical DMAs. In order to fully understand the flows, electric field, and particle trajectories inside a scanning ROMIAC, particle transmission

has been examined with finite-element simulations using COMSOL Multiphysics® (Version 5.3).



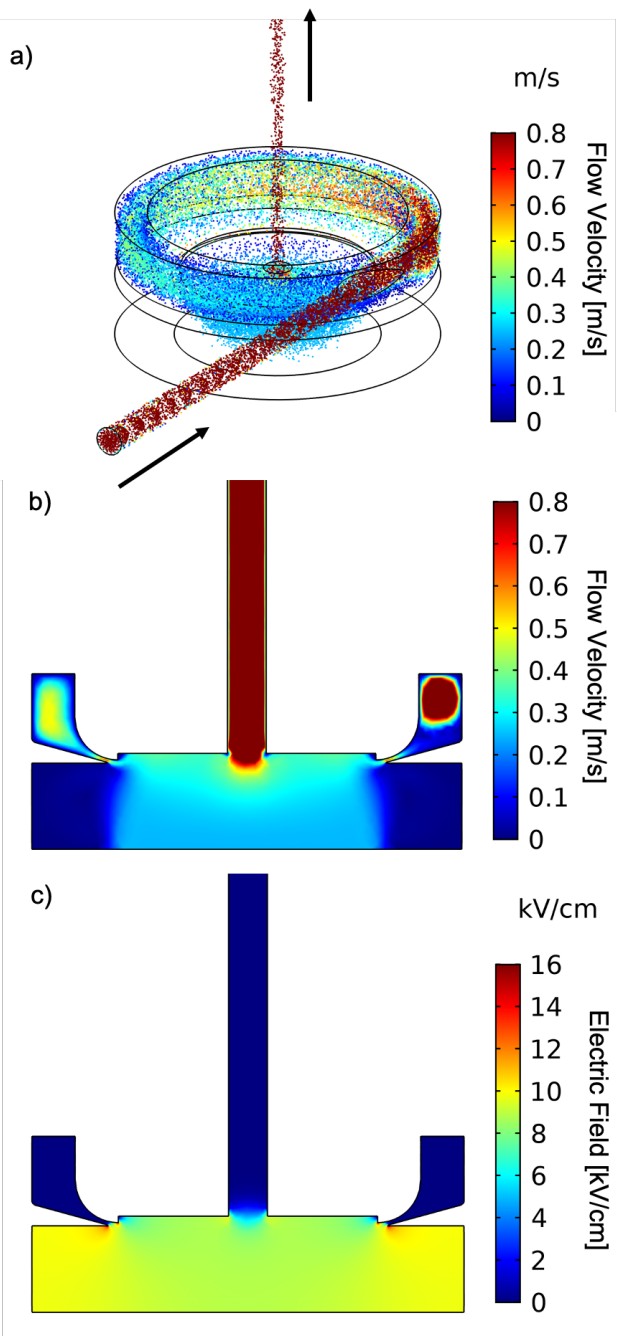

**Figure 2.** Example of finite-element simulations of the scanning ROMIAC at an aerosol flow rate of $Q_a = 1.2$ L/min, a cross flow rate $Q_c = 12$ L/min, with particle diameter of $d_p = 4.0$ nm. (a) Particle trajectories over a 50 s upscan at $t \approx 25$ s. Cross-section view of (b) the flow velocity, and (c) electric field distribution. The magnitude of the electric field corresponds to the maximum, 10 kV, electric potential. Particles enter the ROMIAC from the entrance slit that is tangential to the classification region, and leave the ROMIAC from the slit perpendicular to the classification region.





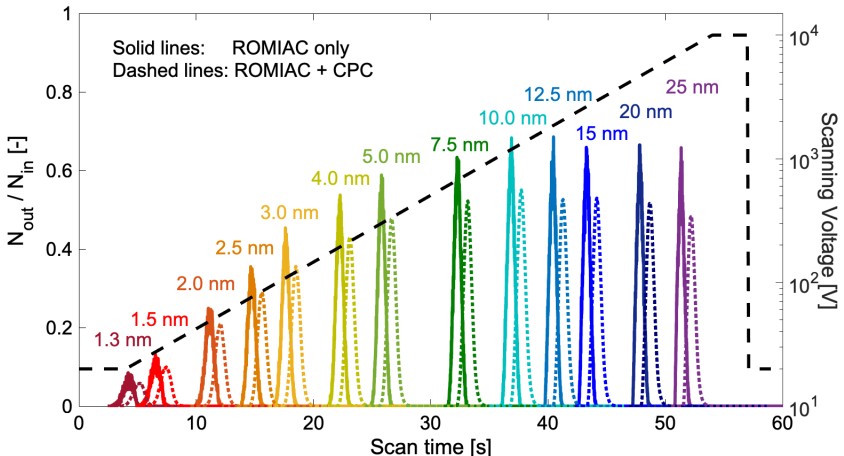

**Figure 3.** Simulated transfer function of the scanning ROMIAC with monodisperse input particles in the $1.3 - 25$ nm size range. The transfer function is calculated as the ratio of particle number at the exit and the entrance of ROMIAC over a voltage scan (dashed black line) with $t_{ramp} = 50$ s. Solid lines show the transfer function of the classifier (scanning ROMIAC) only; dashed lines show the simulated ROMIAC transfer function coupled with the CPC residence time distribution (see Eq. (12)). The integrated transfer function peaks (dashed lines) are used to compute the inversion kernel for nSEMS data inversion.

We have solved Navier-Stokes and Maxwell equations for the flow and electric fields, respectively, using the "Laminar" and "Electrostatic" modules in COMSOL Multiphysics®. The time-varying electric field, $E_{(x,y,z,t)}$, can be treated as quasi-steady-state:

$$E(x,y,z,t) = E_0(x,y,z)f(t) , \qquad (7)$$

where $E_0(x,y,z)$ is the electric field in the beginning of the voltage ramp at $V(t = 0) = V_{low}$, and $f(t)$ is the time variation factor depending on the characteristic ramping time $\tau_s$ defined in Eq.(4):

$$f(t) = \begin{cases} 1, & 0 \leq t < t_{low} \\ e^{\frac{t - t_{low}}{\tau_s}}, & t_{low} \leq t < t_{low} + t_{ramp} \\ \dfrac{V_{high}}{V_{low}}, & t_{low} + t_{ramp} \leq t < t_{low} + t_{ramp} + t_{high} \end{cases} \qquad (8)$$

Because the particles classified in the ROMIAC are sufficiently small that inertial effects can be neglected, particle motion is numerically simulated using the "Particle trajectories" module in COMSOL, with only the drag, electrostatic forces, and 185 Brownian motion being considered, as the particles are assumed to be massless. The scanning ROMIAC transfer function for monodisperse particles can be written as:

$$\Omega_{ROMIAC}(Z_p, \beta, \delta, t) = \frac{N_c(t)}{N_a} , \qquad (9)$$





where $N_a$ and $N_c$ are the number of particles going in and coming out of the ROMIAC during the simulation. In order to determine the scanning ROMIAC transfer function with adequate time resolution, 200 particles are injected into the ROMIAC every 2.5 ms; simulations were performed for the default flows and voltage ramp settings listed in Table 1. The times at which particles enter and exit the scanning ROMIAC were recorded. Figure 2 shows an example of the finite-element solutions of the flow and electric fields for $d_p = 4$ nm particles. The modeled instrument response for particles with 13 different mobility diameters across the sizing range of the instrument are shown with solid lines in Figure 3. The peak transmission ratio for particles larger than about 5 nm remains flat, at approximately 60%, and progressively drops at smaller sizes. Simulation was also performed for different ramp times in order to compare the transfer function distortions that may result from fast voltage scanning (Figure S1).

### 3.2 Laboratory characterization of the ROMIAC

Although particle trajectory simulations using COMSOL Multiphysics® have proven very effective at retrieving particle transfer functions, they cannot fully capture the nonideal, three-dimensional behavior of particles inside the classifier due to the high computation cost (Mai and Flagan, 2018; Amanatidis et al., 2020). As a result, experimental calibrations are needed to closely examine the scanning ROMIAC performance. Figure 4 shows the tandem differential mobility analyzer (TDMA, Rader and McMurry (1986)) calibration setup used; aerosols of a known size are selected with a classifier (a constant voltage ROMIAC or DMA for small, 1 - 20 nm, or large, 12 - 26.5 nm, respectively) before entering the nSEMS. Depending on the target size range, source particles were generated from electrosprayed tetra-alkyl ammonium solutions (Ude and de la Mora, 2005), a heated Nichrome wire, or atomized sodium chloride solution. The polydisperse aerosols generated from the hot-wire or the atomizer were size-selected by a ROMIAC or a cylindrical DMA operating at constant voltage to provide a narrow-mobility distribution sample for nSEMS calibration. In order for the size-selected source particles to approximate a monodisperse aerosol, both the cylindrical DMA and the classifying ROMIAC were run at higher resolution than the nSEMS standard operating condition ($\mathcal{R}_{nd} \geq 10$ for both the classifying DMA and ROMIAC), using open-loop controlled sheath flow or cross flow, respectively.

Due to perturbations of the electric field and imperfections in the instrument fabrication, particle transmission in any mobility analyzer can deviate from the designed performance. When the ROMIAC of the nSEMS is operated in static mode, correction factors can be determined empirically to account for any deviations from theoretical or numerical performance. In terms of particle sizing, an empirical mobility correction factor, $f_z$, is calculated by comparing the experimental transfer function with the expected $Z^*_{p,ideal}$, as defined in Eq.(6) using the TDMA calibration setup (Mui et al., 2017). This correction factor, $f_z = Z^*_p / Z^*_{p,ideal}$, is estimated to be 1.03 for the ROMIAC classifier used in the nSEMS system.

### 3.3 Characterization of the two-stage CPC

In addition to the ROMIAC, nonideal performance of the two-stage CPC may also affect the nSEMS data acquisition and interpretation. Figure 5 shows the experimental setup that was used to measure the size-dependent detection efficiency of the two-stage CPC. The classifying ROMIAC was operated in static mode with a resolving power of $\mathcal{R}_{nd} = 14$. The hot-wire





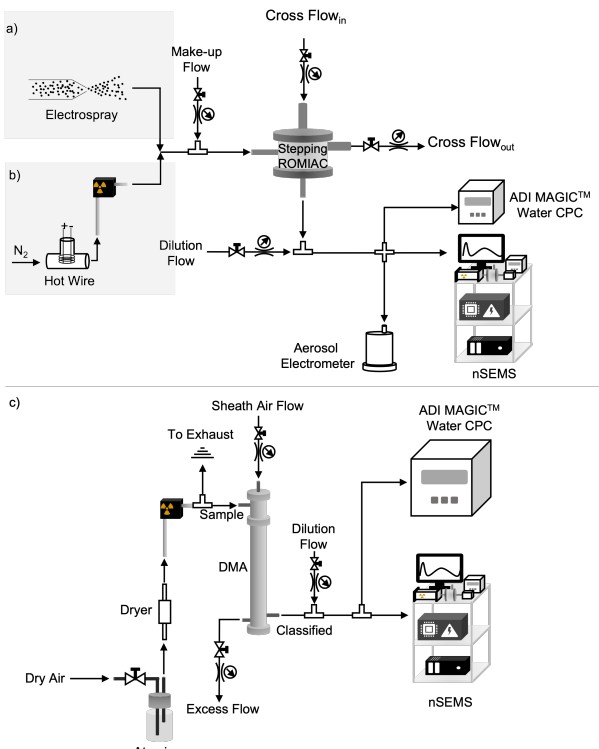

**Figure 4.** The experimental setup used for the nSEMS calibration and characterization in different particle size ranges. (a) $< 3$ nm size range: tetra-alkyl ammonium ions produced by an electrospray were classified using static ROMIAC as classifier ($\mathcal{R}_{nom,nd} \approx 10$), and an aerosol electrometer as a reference for the upstream particle number. The electrospray was operated at 3000 V and 25 cm $H_2O$ pressure. The tetra-alkyl ammonium solutions were prepared with 10 - 20 mg salt in 1.5 ml methanol. (b) $1.5 - 20$ nm size range: A heated Nichrome wire was employed as a hot-wire aerosol source, a static ROMIAC as classifier ($\mathcal{R}_{nom,nd} \approx 10$), and both an aerosol electrometer and ADI MAGIC$^{TM}$ water CPC as upstream particle counters. The hot-wire was operated in the range of 5.0 - 7.0 V and 4.5 - 6.5 A. (c) $12.0 - 26.5$ nm size range: atomized sodium chloride was employed as aerosol source, a cylindrical differential mobility analyzer (DMA) as classifier ($Q_a = 0.5 L/min$, $Q_{sh} = 5.8 L/min$, $\mathcal{R}_{nom,nd} \approx 12$), and an ADI MAGIC$^{TM}$ water CPC as upstream particle counter. Both b) and c) follow a TDMA calibration setup (Rader and McMurry, 1986), which uses a classifier at a constant voltage to select particles within a narrow distribution of sizes.





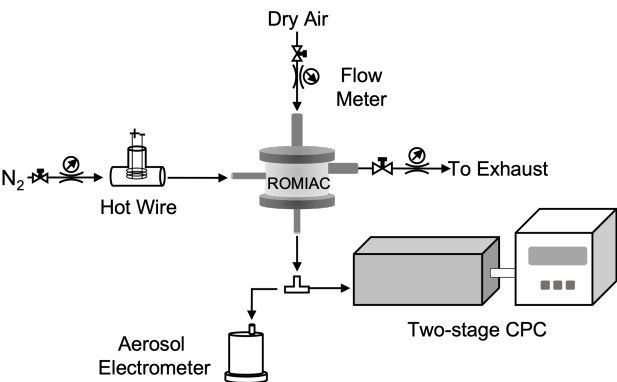

**Figure 5.** Experimental setup used to characterize the two-stage CPC detection efficiency. A heated Nichrome wire (hot-wire) aerosol generator was used to provide aerosol samples. The ROMIAC was operated at static mode to provide stable, monodisperse aerosol particles for both the two-stage CPC and the aerosol electrometer. The ROMIAC aerosol and cross-flow rates were $Q_a = 2.5$ L/min and $Q_x = 35.5$ L/min. The electrometer was pre-calibrated against a TSI 3760A butanol-based CPC and an ADI MAGIC$^{\text{TM}}$ water-based CPC.

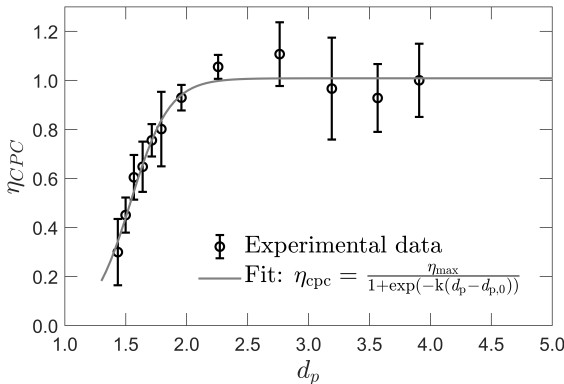

**Figure 6.** Detection efficiency of the two-stage CPC as a function of $d_\text{p}$, the mobility-equivalent particle diameter. The efficiency is corrected for the dilution due to the vapor flow. The size-dependent detection efficiency is fitted to a logistic function with fitting parameters $\eta_\text{max} = 1.01$, $k = 6.30$ nm$^{-1}$, and $d_{\text{p},0} = 1.54$ nm. The fit is used to approximate the CPC detection efficiency, $\eta_\text{cpc}$, in the data inversion.





particle generator was set at similar conditions as shown in Figure 4. Given that all particles are singly charged, an aerosol electrometer was placed in between the ROMIAC and the two-stage CPC to measure the total particle number concentrations coming out of the classifier. The plumbing upstream of the CPC was kept the same as on the integrated nSEMS system so that the resulting plumbing delays would be taken into account in this calibration. The CPC counting efficiency relative to that

of the aerosol electrometer is shown in Figure 6. The 50% cut-off size is about 1.6 nm, and the counting efficiency reaches a plateau when particle mobility-equivalent diameter is larger than about 2.1 nm ($d_p \geq 2.1$nm). The empirically determined two-stage CPC counting efficiency was fitted with a logistic function:

$$\eta_{\mathrm{cpc}} = \frac{\eta_{\max}}{1 + e^{-k(d_{\mathrm{p}} - d_{\mathrm{p},0})}},\tag{10}$$

where $\eta_{\max} = 1.01$, $k = 6.30\ \mathrm{nm}^{-1}$, and $d_{\mathrm{p},0} = 1.54$ nm, based on the calibration results.

In addition to detection efficiency, the delay in CPC response also complicates the transfer function of the system. When the classifier is operated at scanning mode, the slow response of the CPC may introduce a smearing effect and broaden the particle transfer functions. To account for this effect, the response of the two-stage CPC can be modeled as a plug flow reactor (PFR) in series with a continuous stirred-tank reactor (CSTR) to estimate its particle residence time distribution, $E_{\mathrm{cp}}(t)$ (Russell et al., 1995; Collins et al., 2002; Mai et al., 2018):

$$
\begin{aligned}
E_{\mathrm{cp}}(t) &= E_{\mathrm{p}}(t) \star E_{\mathrm{c}}(t) \\
&= \int_{-\infty}^{\infty} E_{\mathrm{c}}(t') E_{\mathrm{p}}(t - t')\mathrm{d}t' \\
&= \begin{cases} 0, & t < \tau_{\mathrm{p}} \\ \frac{1}{\tau_{\mathrm{c}}} e^{-\frac{t-\tau_{\mathrm{p}}}{\tau_{\mathrm{c}}}}, & t \geq \tau_{\mathrm{p}}, \end{cases}
\end{aligned}\tag{11}
$$

where $\tau_{\mathrm{p}}$ and $\tau_{\mathrm{c}}$ are the mean residence time of the PFR and the CSTR, respectively, and $\star$ is the symbol for the convolution of two functions (Bracewell and Bracewell, 1986). The transfer function of the integrated nSEMS system, $\Omega_{\mathrm{nSEMS}}(Z_{\mathrm{p}}, \beta, \delta, t)$,

can be written as

$$\Omega_{\mathrm{nSEMS}}(Z_{\mathrm{p}}, \beta, \delta, t) = \Omega_{\mathrm{ROMIAC}}(Z_{\mathrm{p}}, \beta, \delta, t) \star E_{\mathrm{cp}}(t)\tag{12}$$

  To explore the extent of this effect, the nSEMS was run with different voltage ramp times (10 s - 1400 s, corresponding to a $\tau_s/\tau_f$ value of 4 - 5600). At long voltage ramp times, e.g., at 1400 s, the nSEMS can be treated as operating in a quasi-static mode, where the CPC response time has no impact on the transfer functions. Figure 7 shows the experimentally-determined

particle transfer functions of the nSEMS at different $t_{\mathrm{ramp}}$, for $d_{\mathrm{p}} = 18$ nm particles. The results indicate that the smearing effect is small when $t_{\mathrm{ramp}}$ is longer than 30 s ($\tau_{\mathrm{s}} \geq 4.83$ s). The residence time distribution is computed by deconvoluting the quasi-static nSEMS transfer function measured with $t_{\mathrm{ramp}} = 1400$ s, from that measured with $t_{\mathrm{ramp}} = 50$ s. The resulting characteristic times for the CPC residence time distribution were $\tau_{\mathrm{c}} = 0.20$ s and $\tau_{\mathrm{p}} = 0.70$ s (Figure S2). The dashed lines in Figure 3 show the convoluted nSEMS transfer function, $\Omega_{\mathrm{nSEMS}}(Z_{\mathrm{p}}, \beta, \delta, t)$, combining the CPC residence time distribution in

addition to the scanning ROMIAC simulation.





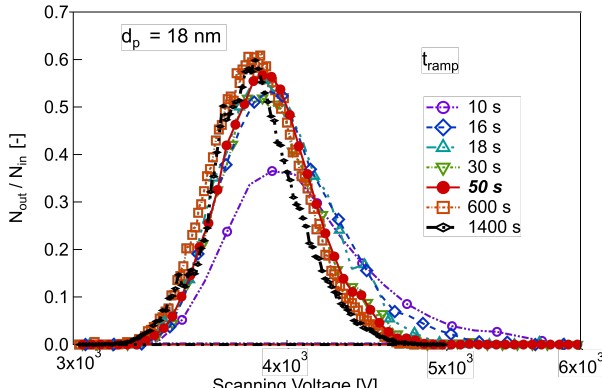

**Figure 7.** Effect of voltage ramp time, $t_{\text{ramp}}$, on the nSEMS scanning transfer function with $d_{\text{p}} = 18$ nm input particles. The nSEMS voltage is increased exponentially from 20 V to 10 kV, over ramp times within $10 - 1400$ s, including the default $t_{\text{ramp}} = 50$ s. CPC smearing of the transfer function increases with decreasing $t_{\text{ramp}}$, and becomes very pronounced at $t_{\text{ramp}} < 20$ s.

### 3.4 Derivation of the integrated instrument transfer function and data inversion

Data inversion is required to retrieve particle size distribution of the source particles, $N$, from the particle counts measured by the CPC, $\mathbf{R}_{\text{nSEMS}}$, given that:

$$\mathbf{R}_{\text{nSEMS}} = \mathbf{A}_{\text{nSEMS}} N \tag{13}$$

where $\mathbf{A}_{\text{nSEMS}}$ is often referred to as the inversion kernel for the instrument. The time-series instrument response can be written as $\mathbf{R}_{\text{nSEMS}} = [R_{\text{nSEMS},1}, R_{\text{nSEMS},2}, \cdots, R_{\text{nSEMS},I}]^{\text{T}}$. With the default nSEMS voltage ramp time, $t_{\text{ramp}} = 50$ s, and the CPC data recording frequency, $t_{\text{c}} = 0.2$ s, the vector length for one complete scan is $I = 250$. The particle number counts recorded by the CPC in the $i^{\text{th}}$ time bin, $R_{\text{nSEMS},i}$, can be represented as the integral of the total number of particles transmitted over the time interval $(i-1)t_{\text{c}} \le t < it_{\text{c}}$:

$$R_{\text{nSEMS},i} = Q_{\text{a}} \int\limits_{(i-1)t_{\text{c}}}^{it_{\text{c}}} \int\limits_{-\infty}^{\infty} N(d_{\text{p}}) \sum_{\phi} p_{\text{charge}}(d_{\text{p}}, \phi) \, \eta_{\text{CPC}}(d_{\text{p}}, \phi) \times \Omega_{\text{nSEMS}}(Z_{\text{p}}(d_{\text{p}}, \phi), \beta, \delta, t) \, \mathrm{d}d_{\text{p}} \, \mathrm{d}t \tag{14}$$

The particle charging probability from the soft x-ray charge conditioner, $p_{\text{charge}}(d_{\text{p}}, \phi)$, was assumed to be that of the Wiedensohler (1988) approximation, and is computed separately for scans of negative and positive polarity. In order to obtain the instrument transfer function, $\Omega_{\text{nSEMS}}(Z_{\text{p}}, \beta, \delta, t)$, for each time bin, the simulated ROMIAC transfer functions were first fitted as a Gaussian function:

$$\Omega_{\text{ROMIAC}}(Z_{\text{p}}, \beta, \delta, t) = a \exp\left(-\frac{(t-b)^2}{2c^2}\right) \tag{15}$$

The three fitting parameters, $a$, $b$, and $c$, were then interpolated over the entire time vector with 250 bins. By substituting the interpolated parameters back into Eq.(15), a ROMIAC transfer function, $\Omega_{\text{ROMIAC}}(Z_{\text{p}}, \beta, \delta, t)$, was generated for each time bin.





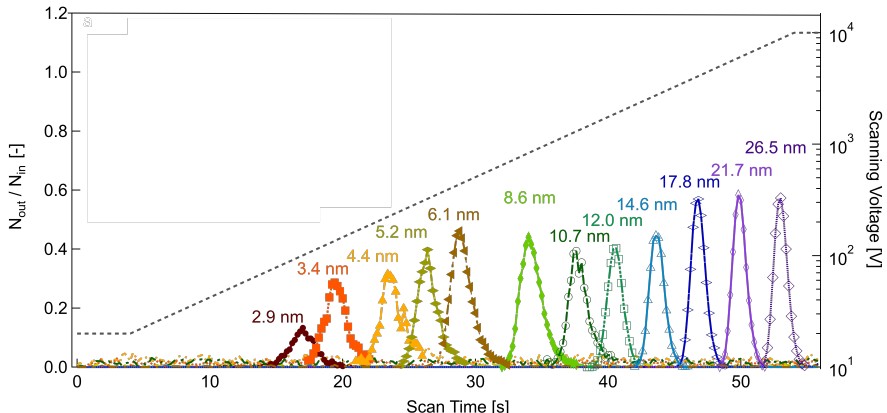

**Figure 8.** Experimental calibration of the nSEMS using the TDMA setup shown in Figure 4. Particles entering the nSEMS were classified as described; the reported mobility-equivalent diameters were calculated based on the upstream classifier operating parameters. The nSEMS was operated at the default parameters listed in Table 1, with $t_{\mathrm{ramp}} = 50\,\mathrm{s}$. The ratio of downstream to upstream particle counts of the nSEMS is shown as a function of time over the voltage scan, with input particles in the $2.9 - 26.5\,\mathrm{nm}$ range. The applied voltage is indicated by the dashed gray line. Only a fraction of the sizes used in the calibration are shown here for clarity; results from the complete size calibration summary are presented in Figure 10.

The fitted transfer functions were adjusted by the empirically determined mobility correction factor, $f_z$. The nSEMS transfer function for each time bin, $\Omega_{\mathrm{nSEMS}}(Z_{\mathrm{p}}, \beta, \delta, t)$, was then computed by the convolution of the ROMIAC transfer function and the CPC residence time distribution of Eq.(12). Similar to the CPC instrument response, the inversion kernel matrix can also be decomposed into a time-series form, $\mathbf{A}_{\mathrm{nSEMS}} = [A_{\mathrm{nSEMS},1}, A_{\mathrm{nSEMS},2}, \cdots, A_{\mathrm{nSEMS},I}]^{\mathrm{T}}$, with the row vector

$$A_{\mathrm{nSEMS},i} = Q_{\mathrm{a}}\, p_{\mathrm{charge}}(d_{\mathrm{p}}, \phi)\, \eta_{\mathrm{CPC}}(d_{\mathrm{p}}, \phi)\, \mathrm{d}d_{\mathrm{p}}\, t_{\mathrm{c}} \times \Omega_{\mathrm{nSEMS},i}(Z_{\mathrm{p}}(d_{\mathrm{p}}, \phi), \beta, \delta, t) \tag{16}$$

We then applied a totally nonnegative least squares (TNNLS) algorithm to retrieve the sample particle size distribution from the inversion kernel and the particle number concentrations detected by the CPC, i.e., solving for $N = \mathbf{A}_{\mathrm{nSEMS}}^{-1}\mathbf{R}_{\mathrm{nSEMS}}$ (Merritt and Zhang, 2005).

### 3.5 Calibration results

Figure 8 shows the nSEMS scanning response with size-selected particles of different mobility-equivalent diameters using the TDMA setup shown in Figure 4b ($d_{\mathrm{p}} \leq 10.70\,\mathrm{nm}$) and Figure 4c ($d_{\mathrm{p}} \geq 12.00\,\mathrm{nm}$) with the default operating parameters listed in Table 1. The particles fed into the nSEMS using the TDMA set-up were not monodisperse, which may have led to peak broadening and particle losses. In order to compare the experimental result with simulation, we deconvoluted the CPC residence time distribution and the static classifier transfer function from the instrument response as shown in Figure 8 (Stolzenburg and



McMurry, 2008):

$$\Omega_{\mathrm{ROMIAC}}(\tilde{Z}_\mathrm{p}, \beta, \delta, t) = \begin{cases} \dfrac{R_{\mathrm{nSEMS},i}}{E_{\mathrm{cp}}\Omega_{\mathrm{ROMIAC}}(\tilde{Z}_\mathrm{p}, \beta, \delta, \sigma)}, & d_\mathrm{p} < 12\mathrm{nm} \\[2ex] \dfrac{R_{\mathrm{nSEMS},i}}{E_{\mathrm{cp}}\Omega_{\mathrm{DMA}}(\tilde{Z}_\mathrm{p}, \beta, \delta, \sigma)}, & d_\mathrm{p} \geq 12\mathrm{nm} \end{cases} \tag{17}$$

where the dimensionless mobility, $\tilde{Z}_\mathrm{p}$, is defined as the ratio of input particle electrical mobility and the ideal mobility, $Z^*_{\mathrm{p,ideal}}$,

as defined in Eq. 6:

$$\tilde{Z}_\mathrm{p} = \frac{Z_\mathrm{p}}{Z^*_{\mathrm{p,ideal}}} \tag{18}$$

and $\Omega_{\mathrm{ROMIAC}}(\tilde{Z}_\mathrm{p}, \beta, \delta, \tilde{\sigma})$ and $\Omega_{\mathrm{DMA}}(\tilde{Z}_\mathrm{p}, \beta, \delta, \tilde{\sigma})$ are the diffusing Stolzenburg transfer function for the ROMIAC and long-column DMA operated at static mode, respectively:

$$\begin{aligned} \Omega_{\mathrm{ROMIAC,DMA}}(\tilde{Z}_\mathrm{p}, \beta, \delta, \tilde{\sigma}) \;=\; & \frac{\tilde{\sigma}}{\sqrt{2}\beta(1-\delta)} \left[ \epsilon\left(\frac{\tilde{Z}_\mathrm{p} - (1+\beta)}{\sqrt{2}\tilde{\sigma}}\right) + \epsilon\left(\frac{\tilde{Z}_\mathrm{p} - (1-\beta)}{\sqrt{2}\tilde{\sigma}}\right) \right. \\ & \left. - \; \epsilon\left(\frac{\tilde{Z}_\mathrm{p} - (1+\beta\delta)}{\sqrt{2}\tilde{\sigma}}\right) - \epsilon\left(\frac{\tilde{Z}_\mathrm{p} - (1-\beta\delta)}{\sqrt{2}\tilde{\sigma}}\right) \right] \end{aligned} \tag{19}$$

where $\epsilon$ is:

$$\epsilon(x) = x\,\mathrm{erf}(x) + \frac{\exp\left(-x^2\right)}{\sqrt{\pi}} \tag{20}$$

and $\mathrm{erf}(x)$ is the error function. The dimensionless diffusion factor, $\tilde{\sigma}$, is defined as:

$$\tilde{\sigma}^2 = \frac{G_{\mathrm{class}}}{\mathrm{Pe}_{\mathrm{mig}}}\tilde{Z}_\mathrm{p} \tag{21}$$

At ambient temperature, the migration Péclet number for singly-charged particles can be approximated as a function of the static voltage:

$$\mathrm{Pe}_{\mathrm{mig}} = \frac{\phi e V}{kT} \approx \frac{V}{0.0255[V]} \tag{22}$$

The dimensionless geometry factor for classifiers, $G_{\mathrm{class}}$, is estimated to be $G_{\mathrm{LDMA}} = 2.55$ for the TSI 3081 LDMA at $\mathcal{R}_{\mathrm{nd,\,DMA}} \approx 12$, and for the ROMIAC at $\mathcal{R}_{\mathrm{nd,\,ROMIAC}} \approx 10$, can be computed as:

$$G = \begin{cases} \frac{8}{3}, & \xi = 0 \\[2ex] \dfrac{4\left\{ \frac{4}{15}\left[(1-|\xi|^{5/2}) - (1-|\xi|)^{5/2}\right] + \frac{1}{3}\left(\frac{\xi}{\alpha}\right)^2\left[(1-|\xi|^{3/2}) - (1-|\xi|)^{3/2}\right] \right\}}{|\xi|(1-|\xi|)}, & 0 < |\xi| < 1 \\[2ex] 2\left[\frac{4}{3} + \left(\frac{1}{\alpha}\right)^2\right], & |\xi| = 1 \end{cases} \tag{23}$$

where $\xi = \beta^{-1}(\tilde{Z}_\mathrm{p} - 1)$ and $\alpha = L/b = 0.015\ \mathrm{m}/0.01\ \mathrm{m} = 1.5$ for the ROMIAC. The real resolution of the scanning ROMIAC can then be computed using the full width at half maximum of the transfer function, $\Omega_{\mathrm{ROMIAC}}(\tilde{Z}_\mathrm{p}, \beta, \delta, t)$, (Flagan, 1999, 2004):

$$\mathcal{R} = \frac{Z^*_\mathrm{p}}{\Delta Z_{\mathrm{p,FWHM}}} \tag{24}$$

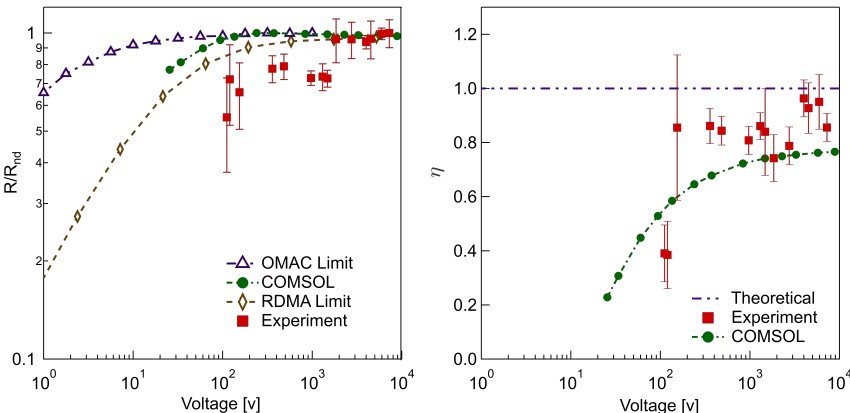

**Figure 9.** Comparison between experimental, simulated, and theoretical transfer functions. (a) Effect of operating voltage on classifier resolution, calculated as the actual resolution based on the full width at half maximum (FWHM) (Eq. 24) over $\mathcal{R}_{\text{nd}}$, the resolution at the nondiffusive regime (>5000V). (b) Particle transmission efficiency as a function of operating voltage. Transmission efficiency, $\eta$, is calculated as the ratio of the actual over the ideal area below the transfer function peak. The error bars represent one standard deviation of uncertainty from multiple experiments at one size.

The transmission efficiency can be calculated as the area under the transfer function normalized by the area of an ideal transfer function, which in non-dimensionless form is equivalent to the flow factor, $\beta$:

$$\eta = \frac{1}{\beta} \int \Omega_{\text{ROMIAC}}(\tilde{Z}_{\text{p}}, \beta, \delta, t) \, \mathrm{d}\tilde{Z}_{\text{p}} = \frac{1}{\beta} \int \frac{N_{\text{out}}(t)}{N_{\text{in}}} \, \mathrm{d}t \tag{25}$$

Figure 9 shows the comparison of ROMIAC resolution and efficiency between experiment, COMSOL simulation, and the theoretical limit calculated for the DMA and the OMAC operated in static mode (Flagan, 2004). The overall performance of

the nSEMS shows convincing agreement with the finite-element simulation results in Figure 3, which proves the feasibility of coupling laboratory calibrations with numerical simulation to predict the instrument response of a SEMS system. Compared to the simulation and theoretical calculation, the effect of diffusional degradation at low voltages remains minimal for the scanning ROMIAC system compared to other conventional nano-SMPS systems, as previously predicted for the static OMAC (Flagan, 2004; Downard et al., 2011). Figure 10 shows the nSEMS peak voltage ($V^*$) as a function of particle mobility when

operated at a cross-flow rate of $Q_c = 12 \, \text{L/min}$. The relatively high classification voltage (~35 V at 1.47 nm) compared to DMAs further reduces diffusional degradation of the resolution (Flagan, 1999). In addition to the calibration results using the hot-wire or atomized particle sources, it also includes the signature peak of tetra-heptyl ammonium bromide (THAB) ions (see Figure 4a for setup). Particle mobilities are calculated using Eq.(1) at given diameters, assuming that the particles are singly charged. The experimentally-determined voltages at the transfer-function peaks are in close agreement with those predicted

by the COMSOL simulation in Figure 3. From the laboratory characterization results, the method of using empirical data to adjust the simulated particle transmission has proven to be an efficient and effective way to derive SEMS or SMPS system transfer function. In addition, the nSEMS, as the first system that employs an opposed-migration classifier with continuously


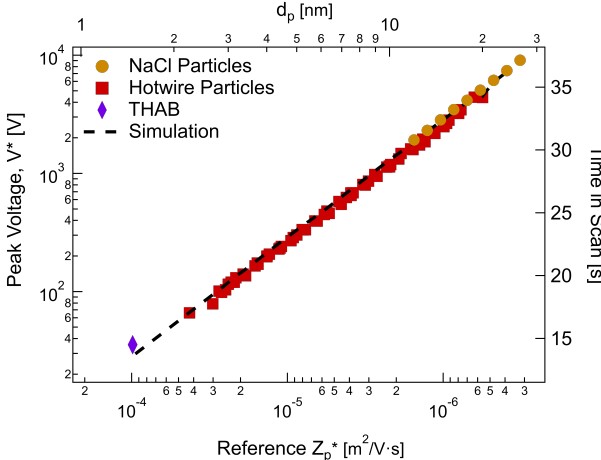

**Figure 10.** nSEMS voltage at the peak transmission ($V^*$) as a function of the input reference particle mobility, $Z_p^*$. Symbols represent experimental results with tetra-heptyl ammonium bromide (THAB), hot-wire, and atomized NaCl particles produced using the setups shown in Figure 4. The black dotted line demonstrates the voltage at peak transmission predicted by the finite-element simulations. The top axis shows the corresponding particle diameter, $d_p$, at a given mobility, $Z_p^*$, as defined in Eq.(1); the right axis shows the corresponding time in the scan.

varying voltage, has also demonstrated the great potential of scanning OMAC systems for providing fast and accurate particle size information in the low-nanometer regime without significant diffusional degradation.

## 4  Application to particle size distribution measurement

Atmospheric new particle formation and its subsequent growth have a great impact on aerosol number concentrations and the Earth's total energy budget. In order to better understand the climate significance of NPF, much research has attempted to study the mechanisms of nucleation and the growth rates of nanoparticles. For example, the CLOUD experiments at CERN have extensively probed the roles of sulfuric acid, ammonia, cosmic rays, and other atmospheric components on nucleation (Kirkby et al., 2011, 2016). To determine the particle formation and growth rates from the atmospheric chamber experiments, the particle size distribution needs to be measured at high resolving power and with a short duty cycle. In addition to being able to capture the transient aerosol dynamics during NPF events, since most of the nucleation occurs in clean atmospheric conditions, the instrument must be capable of taking measurements at relatively low particle concentrations. The scan rate selected for this initial implementation was, therefore, a compromise between fast response and counting statistics.

The nSEMS was used with a 60-second scan for particle sizing in both the CLOUD 13 and the CLOUD 14 campaigns. Figure 11 shows a particle size distribution measured with nSEMS during an ion-induced nucleation event that simulated atmospheric nucleation and nanoparticle growth in urban environment in CLOUD 13. The experiment was conducted in the presence of sulfuric acid, nitric acid, and ammonia at -10 °C and 60 % RH. When particles reached $d_p \approx 4.6$ nm, nitric acid



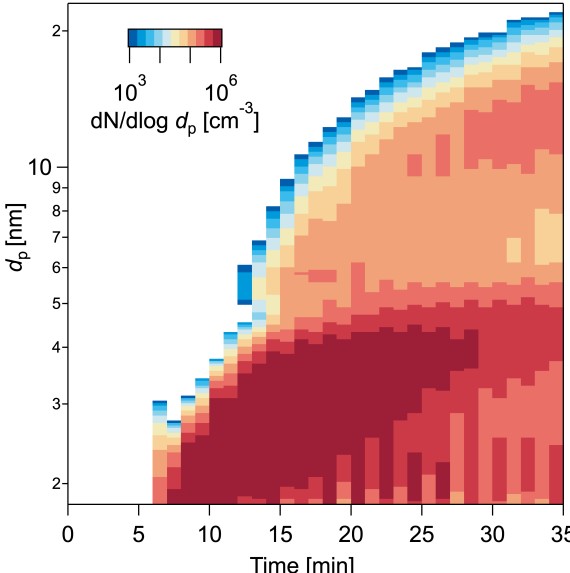

**Figure 11.** Particle size distribution measured by the nSEMS during a nucleation and growth event in the CLOUD 13 campaign with anthropogenic trace gases. The experiment was conducted at -10°C and 60% RH, with 24 pptv $HNO_3$, 2131 pptv $NH_3$, 0.46 pptv $H_2SO_4$, and 0.28 pptv highly oxygenated organic molecules (HOM). The nSEMS high voltage polarity was switched between scans to probe both positively and negatively charged particles from the soft x-ray charge conditioner. A clear bimodal size distribution was observed by the nSEMS due to the rapid co-condensation of nitric acid and ammonia (Wang et al., 2020). The activation diameter, $d_{act}$, for nitric acid condensation is around 4.6 nm.

and ammonia started condensing rapidly onto the particles, resulting in a growth rate of 40 nm/h. This extremely fast growth from nitric acid and ammonia co-condensation can generally persist for only a few minutes, and activate only the largest of the initial small nuclei, before depleting the nitric acid supersaturation and turning off additional nucleation. Those few nuclei that activate are often present only in low concentrations (Wang et al., 2020). Consequently, the other conventional particle sizing instruments that are connected to the CLOUD chamber were not able to fully capture this rapid growth event; the concentrations were too low to be detected by the nano-SMPS (Tröstl et al., 2015). Moreover, the size of the particles

evolved so fast that a higher size resolution was required than could be attained by the DMA-train that measures six sizes in parallel with separate static DMAs (Stolzenburg et al., 2017). In the region where multiple instruments can capture the aerosol dynamics, the intercomparison between the results of different instruments showed reasonably good agreement. Combining these measurements also provided detailed particle growth information in rapidly evolving new particle formation events that the other instruments could follow Wang et al. (2020). In addition to enabling high resolving power measurements of size

distributions during rapid particle growth events, the nSEMS also provides valuable information on natural ion and charged particle concentrations in the chamber when operated with the charge conditioner switched off. The ability to measure the





concentrations of positive and negative nanoparticles separately facilitates study of the role of ions in atmospheric nucleation and growth.

## 5  Conclusions

The design and performance of a novel nanoparticle size-classifying instrument, the nSEMS, has been evaluated. The concept of OMAC was first proposed in order to overcome the diffusional degradation at lower voltages of the DMA (Flagan, 2004). The radial form of the OMAC, the ROMIAC, was then designed to classify nanoparticles in the low-nanometer regime with high resolving power in static mode (Mui et al., 2013, 2017). According to the ideal model of OMAC, particles are transmitted through the classification region parallel to the porous electrodes, and voltage variations would lead to excessive particle losses.

A key feature of the ROMIAC design was to both introduce the sample and to extract the classified ions or particles on the ground-electrode side of the classifier. The resulting trajectories, which can be seen in COMSOL Multiphysics$^{\text{TM}}$ simulations shown in Figure S1, reduce losses associated with voltage scanning to acceptable levels, thereby enabling measurement acceleration by voltage scanning and operating as a SEMS. The ability to classify particles at low voltage with minimal diffusional degradation of the transfer function, combined with a fast response CPC that minimizes residence time distribution related to

the smearing effect, made it possible for the ROMIAC to be operated with fast exponential voltage ramping, greatly accelerating the measurement over that of static-mode operation. The nSEMS system, which uses a soft x-ray charge conditioner, a scanning ROMIAC as classifier, and a two-stage CPC as particle detector, can provide highly resolved particle size distribution measurements, in the 1.5 - 25 nm size range in one minute or less (we did not push the bound on the scan rate in the initial application of the nSEMS at CLOUD). The integrated instrument transfer function, which can reproduce how particles are

transmitted inside the nSEMS within 10% uncertainty, has been derived by combining COMSOL finite-element analysis with empirical adjustments. The particle size distributions measured by the nSEMS employing the described data inversion method agrees reasonably well with other instruments (Tröstl et al., 2015; Stolzenburg et al., 2017) used in the CLOUD experiment (Wang et al., 2020). However, there remain uncertainties associated with particle charge distribution. The dual-polarity scanning feature of the nSEMS makes it possible to observe the charge effects on the evolution of the size distribution as particles

nucleate and grow. Its dual polarity capability should also facilitate characterization of the particle charge distribution in the low nanometer regime, thereby improving the instrumental transfer function and data inversion. Overall, this instrument is able to provide robust particle sizing information in the sub-25 nm region, and is extremely powerful in examining atmospheric nucleation and the subsequent growth of nanoparticles.

*Code and data availability.*  Data and code related to this article are available upon request to the corresponding author.



*Author contributions.* W.K. wrote the paper, analyzed the data, led the calibration of the instrument, operated the instrument at CLOUD. S.A. led the construction the instrument as deployed to CLOUD, developed the data acquisition and control software, contributed to the calibration of the instrument, and contributed to the writing of the manuscript. H.M. built and conducted proof-of-concept experiments with the first prototype of the instrument, and developed the finite element simulation in developing the instrument response function. C.K. developed the soft x-ray charger used in the apparatus, and did the initial construction of the two-stage condensation particle counter detector. B.C.S

contributed to the construction and calibration of the instrument, and operated it in CLOUD experiments. Y.H. contributed to the development of the data analysis software and contributed to the writing of the manuscript. G.S.L. developed the modified MAGIC CPC used as the second stage of the detector. S.V.H. led the modifications of the MAGIC CPC for use in this instrument. J.H.S. contributed to the analysis of the data, and the writing of the manuscript. R.C.F. led the project, development of the instrument, design of the experiments, and data analysis, along with contributing to the writing and editing of the manuscript.

*Competing interests.* The California Institute of Technology has a patent pending for the nSEMS. Aerosol Dynamics, Inc., has a patent on the MAGIC CPC. No other conflicts of interest were identified.

*Acknowledgements.* The authors would like to thank the CLOUD experiment for providing the facility for instrument testing and operation. We gratefully acknowledge a generous gift from Christine and Dwight Landis to improve the data acquisition system of the instrument. This work was supported by the National Science Foundation under Grants No. AGS 1602086 and AGS 1801329.



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
