# Peer review of "The nano-scanning electrical mobility spectrometer (nSEMS) and its application to size distribution measurements of 1.5-25 nm particles"

_Atmospheric Measurement Techniques, 2021_

## Referee Comment (RC1)

**The nano-scanning electrical mobility spectrometer (nSEMS) and its application to size distribution measurements of 1.5-25 nm particles** by Weimeng Kong, Stavros Amanatidis, Huajun Mai, Changhyuk Kim, Benjamin C. Schulze, Yuanlong Huang, Gregory S. Lewis, Susanne V. Hering, John H. Seinfeld, and Richard C. Flagan

This study describes in considerable detail a powerful instrument that has already produced valuable experimental data on new particle formation in the carefully controlled CLOUD experiment (Wang et al. 2020). The instrument is able to generate with fine sensitivity and resolution a size distribution in the 1.5-25 nm range in 60 s, which is apparently faster than previously achieved. Additionally, the article provides extensive information on design criteria, which should be most valuable in future studies aimed at making comparable measurements with improved performance. The experimental and interpretive challenges overcome are great in many fundamental dimensions, including sampling, charging, transmission, size resolution, sensitivity of detection, speed of measurement, etc. Handling all these problems in a single instrument has required a difficult strategic thinking on components choices, and this is also carefully discussed with extensive knowledge of related literature. Globally considered this is an outstanding contribution to the field of atmospheric measurement, which should by all means be published. The choice of journal is also excellent, as the article provides an ideal venue for discussion of the many challenges involved in these measurements, as well as on the many possible alternative experimental tools that were not chosen.

In my discussion I will focus on these other alternatives. I realize that the authors have contributed their own practical solution to their global measurement problem, including a fair amount of justification for their strategic choices. They have accordingly no duty to engage in the more extensive discussion I propose. Nevertheless, given their wide experience, their response would be exceedingly useful to the many colleagues familiar with the various components of this instrument, but not necessarily with their optimal integration into a functioning system.

1. The two-stage condensation detector is justified as follows: *Some single-stage CPCs have been operated at sufficiently high supersaturation to activate particles as small as 1 nm diameters, but in the experiment for which this instrument has been developed, where measurements must be made in a high-radiation environment, this can lead to nucleation within the CPC. Therefore, we took a more conservative approach that has proven robust and effective for sub-10nm particle detection, namely a two-stage CPC, in which the first stage employs a low vapor pressure working fluid, typically diethylene glycol (DEG) that can activate small particles with minimal risk of homogeneous nucleation (Iida et al., 2009).*

Assuming that the nucleation region cannot be shielded, the effect of radiation would be to create small ions in air, similar to those produced in the X-ray chamber, but not removable by the DMA. One must then select a vapor able to discriminate between small ions produced by radiation and 1.5 nm particles. But why would this be precluded in common CPC vapors? In theory (say in Thomson's classical model for nucleation on charged particles) all vapors have critical activation curves that depend on size, which theoretically enables the exclusion of small air ions. Furthermore, Tauber et al. (ChemPhysChem 2018, 19, 3144– 3149) have recently found significant differences between the critical supersaturation for various atomic ions in butanol vapors. Similarly, Attoui and colleagues (doi: https://doi.org/10.1016/j.jaerosci.2021.105772) report that Kanomax's fast CPC can detect 2 nm particles. My point is that there may be single CPC solutions to the detection problem that, in addition to being simpler, would be faster. If these

solutions are presently unknown or insufficiently reliable, please state so, as this would provide a healthy stimulus for developing them.

2. DMA selection. The selection of the DMA used is justified because "*radial opposed migration ion and aerosol classifier (ROMIAC), can classify nanometer-sized particles with minimal degradation of its resolution or diffusional losses.* Another advantage of ROMIAC noted is that it achieves resolving powers comparable to those of conventional DMAs, though at substantially reduced flow rates. Nevertheless, given that a key advance in the new instrument is to reduce measurement time, it would seem that a larger gas flow rate would be preferable. For instance, the Half-Mini DMA has a residence time typically below 1 ms in the analyzer.[1] Another important issue we have recently noted is that high resolution is directly relevant to measurement speed because the residence time in the analyzer is almost identical for all ions of a given mobility, and therefore appears as a pure delay easily corrected in a fast measurement.[2] In that study we argue that a full mobility scan can be completed within a few seconds by combining a detector with a response faster that 20 ms with the half-mini DMA. Although such a fast CPC exists, that claim remains to be experimentally substantiated.

Figure 8 of the article shows that the resolution of the ROMIAC is excellent even for the sophisticated nucleation studies pursued. However, the figure also shows that the width of the spectrum of all monomobile particles used spans several seconds in the 60 s temporal spectrum. A DMA of higher resolution would not only resolve perfectly the 3.4 and 2.9 particles (partially overlapping in Figure 8). Even more usefully, much narrower peaks would enable much faster scans without loss of resolution.

One could be led to conclude from the nucleation event illustrated in Figure 11 that a temporal resolution of 1 minute is adequate. This may be true for events lasting several minutes, but perhaps not in situations where the reservoir of vapor available for particle growth is more limited. Given that the improved time resolution of the instrument of Kong et al. enabled the capture of nucleation events that would have been missed with slower prior equipment, one may surmise that currently undetectable events will be capturable by future devices having more temporal resolution.

There is another potential advantage of higher resolution in nucleation studies, as well as in other situations with a natural mechanism to produce narrow size distributions. If the signal is concentrated into a narrow range of sizes, being able to resolve them would increase the signal. In Figure 8 of Kong et al. one sees that the peak corresponding to 3.4 nm particles spans a size range from 2.9 to about 3.9 nm. The atmospheric nucleation spectrum shown in Figure 11 includes mobility peaks of comparable width, suggesting that perhaps the real size distribution along the growth curve is considerably narrower and taller than can be captured with the available resolving power. In other words, higher resolution would enable detecting lower intensity as well as shorter nucleation events. It would also provide a more precise measurement of the growth rate.
* * *
[1] J. Fernandez de la Mora, Expanded flow rate range of high-resolution nanoDMAs via improved sample flow injection at the aerosol inlet slit, J. Aerosol Sci. 113, 265-275, 2017
[2] J. Fernandez de la Mora, L.J. Perez-Lorenzo, G. Arranz, M. Amo-Gonzalez, H. Burtscher, Fast high-resolution nanoDMA measurements with a 25 ms response time electrometer, Aerosol Science and Tech., 51(6), 724 – 734, 2017

3. Soft X-ray ionizer. This element does not receive as detailed a description as other components of the system. The choice of a bipolar source is defended based on the freedom it offers to examine positive and negative polarities. This is indeed a most useful feature, especially for particles of such small sizes, whose charging probability has not been yet well studied. Soft X-rays have an evident regulatory advantage over radioactive sources. However, all ionization sources relying on energetic particles pose a danger of converting organic volatiles into involatile species, causing artificial nucleation events in sufficiently polluted atmospheres. The details of how this tendency is moderated by Kong et al. would be of interest to the reader. In this realm, it is worth noting that the advent of electrospray ionization did change this situation long ago in the case of unipolar sources. One relevant feature of electrospray ionization at atmospheric pressure is that, in contrast to electrical discharges, ions are created under strictly thermal conditions. Vapors are certainly introduced through the evaporation of a solvent. However, in high conductivity electrosprays, solvent flow rates may be as small as $10^{-7}$ g/s. For this reason, electrosprays of pure volatile solvents containing volatile salts such as ammonium acetate have had an increased use in chemical analysis as a clean and efficient ionization source for vapors. [3] More recently, bipolar electrospray sources combining a positive and a negative emitter [4] have provided a clean substitute for bipolar sources based on ionizing radiation. There is some literature on the ionization probability of unipolar ES sources with vapors, and some untested calculations of how this probability would depend on particle size.[3] We are not aware of studies on the size dependence of the charging probability of nanoparticles in bipolar ES sources, but a similar ambiguity exists for bipolar sources of ionizing radiation. Hopefully these two gaps will be filled soon.

Juan Fernandez de la Mora
Juan.delamora@yale.edu

[3] J. Fernandez de la Mora, Ionization of vapor molecules by an electrospray cloud, International J. Mass Spectrom., **300** 182-193 (2011)

[4] Fernandez de la Mora, J., Barrios, C. (2017) A Bipolar electrospray source of singly charged salt clusters of precisely controlled composition, Aerosol Science and Technology, 51(6) 778 – 786, 2017

---

## Author Comment (AC1)

Reply to Reviewer 1, Juan Fernandez de la Mora:

*This study describes in considerable detail a powerful instrument that has already produced valuable experimental data on new particle formation in the carefully controlled CLOUD experiment (Wang et al. 2020). The instrument is able to generate with fine sensitivity and resolution a size distribution in the 1.5-25 nm range in 60 s, which is apparently faster than previously achieved. Additionally, the article provides extensive information on design criteria, which should be most valuable in future studies aimed at making comparable measurements with improved performance. The experimental and interpretive challenges overcome are great in many fundamental dimensions, including sampling, charging, transmission, size resolution, sensitivity of detection, speed of measurement, etc. Handling all these problems in a single instrument has required a difficult strategic thinking on components choices, and this is also carefully discussed with extensive knowledge of related literature. Globally considered this is an outstanding contribution to the field of atmospheric measurement, which should by all means be published. The choice of journal is also excellent, as the article provides an ideal venue for discussion of the many challenges involved in these measurements, as well as on the many possible alternative experimental tools that were not chosen.*

*In my discussion I will focus on these other alternatives. I realize that the authors have contributed their own practical solution to their global measurement problem, including a fair amount of justification for their strategic choices. They have accordingly no duty to engage in the more extensive discussion I propose. Nevertheless, given their wide experience, their response would be exceedingly useful to the many colleagues familiar with the various components of this instrument, but not necessarily with their optimal integration into a functioning system.*

Response:

We thank the reviewer for his kind words.  He correctly notes in his opening comments that the design of an instrument and the interpretation of the data that it generates has posed many challenges.  Indeed, these challenges introduced conflicting demands on its design, and the specification of its operating conditions.  The objectives of this design included measurement ranging from clusters of sizes probed by mass spectrometers to stable particles that can continue to grow to become cloud condensation nuclei with the time and size resolution needed to quantify the dynamics of these new particles.  The questions raised are constructive points about alternate choices that could have been made.  We note that the instrument on which we report was designed measurements either in the atmosphere or in chamber studies that simulate the atmosphere.  Thus, we address those suggestions in light of the experiments for which this instrument was designed.

As a general point on our responses, we note that the primary focus, and unique feature of the instrument that we report, the nano scanning electrical mobility spectrometer (nSEMS), is the voltage-scanning radial opposed migration ion and aerosol classifier (ROMIAC).  The measurement system comprising the nSEMS must include a detector and a charge conditioner; other designs of those components of the integrated instrument, could be substituted for the ones that we report without changing the essential features of the nSEMS.

We feel that the new capability enabled by the scanning ROMIAC warrants publication of this paper; some of the questions raised will be addressed in future papers.

1. The two-stage condensation detector is justified as follows: *Some single-stage CPCs have been operated at sufficiently high supersaturation to activate particles as small as 1 nm diameters, but in the experiment for which this instrument has been developed, where measurements must be made in a high-radiation environment, this can lead to nucleation within the CPC. Therefore, we took a more conservative approach that has proven robust and effective for sub-10nm particle detection, namely a two-stage CPC, in which the first stage employs a low vapor pressure working fluid, typically diethylene glycol (DEG) that can activate small particles with minimal risk of homogeneous nucleation (Iida et al., 2009).* Assuming that the nucleation region cannot be shielded, the effect of radiation would be to create small ions in air, similar to those produced in the X-ray chamber, but not removable by the DMA. One must then select a vapor able to discriminate between small ions produced by radiation and 1.5 nm particles. But why would this be precluded in common CPC vapors? In theory (say in Thomson's classical model for nucleation on charged particles) all vapors have critical activation curves that depend on size, which theoretically enables the exclusion of small air ions. Furthermore, Tauber et al. (ChemPhysChem 2018, 19, 3144– 3149) have recently found significant differences between the critical supersaturation for various atomic ions in butanol vapors. Similarly, Attoui and colleagues (doi: https://doi.org/10.1016/j.jaerosci.2021.105772) report that Kanomax's fast CPC can detect 2 nm particles. My point is that there may be single CPC solutions to the detection problem that, in addition to being simpler, would be faster. If these solutions are presently unknown or insufficiently reliable, please state so, as this would provide a healthy stimulus for developing them.

Response:

A single-stage condensation particle counter (CPC) detector could be used instead of the two-stage CPC that has been employed in the nano scanning electrical mobility spectrometer (nSEMS).  The reviewer  is correct in noting that the two-stage CPC slows the instrument response since each of the activation and growth stages within the two-stage CPC has its own finite time response.  A number of investigators have demonstrated single-stage CPCs that can detect particles approaching 1 nm in size, and several working fluids have been shown to work (e.g., butanol and diethylene glycol: Kuang et al., 2012, Aerosol Sci Tech 46: 309–15; water: Hering et al., 2017, Aerosol Sci Tech 51: 354–62).  For many applications, these instruments would perform well.

The CLOUD experiment at CERN, where nSEMS was first deployed, employs a 3 GeV pion beam to generate ion concentrations comparable to those in the upper troposphere.  The instrument cannot be shielded from the pions.  Ion generation within the supersaturated volume of the CPC led to false counts during beam events in some early experiments.  In the two-stage CPC, the initial activation is achieved in a high supersaturation, but the rate of nucleation within the instrument is constrained by the high surface tension and relatively low vapor pressure of diethylene glycol.  Growth to optically detectable size is achieved with a lower supersaturation of a more volatile working fluid, minimizing the risk of nucleation within the CPC and the associated false counts, even if ions are generated.   We, therefore, took the conservative approach of using the two-stage CPC as the detector for the experiments for which the nanoSEMS was designed,  in spite of its slower response than the single-stage CPC.

We did, however, replace the TSI butanol CPC that we employed as the second stage in initial experiments with the nSEMS with a faster response, water-based, Aerosol Dynamics MAGIC CPC to speed the detector response. As the reviewer suggests, other experiments may benefit from the faster response of the single-stage CPCs.

2. DMA selection. The selection of the DMA used is justified because "*radial opposed migration ion and aerosol classifier (ROMIAC), can classify nanometer-sized particles with minimal degradation of its resolution or diffusional losses.* Another advantage of ROMIAC noted is that it achieves resolving powers comparable to those of conventional DMAs, though at substantially reduced flow rates. Nevertheless, given that a key advance in the new instrument is to reduce measurement time, it would seem that a larger gas flow rate would be preferable. For instance, the Half-Mini DMA has a residence time typically below 1 ms in the analyzer. Anotherimportant issue we have recently noted is that high resolution is directly relevant to measurement speed because the residence time in the analyzer is almost identical for all ions of a given mobility, and therefore appears as a pure delay easily corrected in a fast measurement. study we argue that a full mobility scan can be completed within a few seconds by combining a detector with a response faster that 20 ms with the half-mini DMA. Although such a fast CPC exists, that claim remains to be experimentally substantiated.

Figure 8 of the article shows that the resolution of the ROMIAC is excellent even for the sophisticated nucleation studies pursued. However, the figure also shows that the width of the spectrum of all monomobile particles used spans several seconds in the 60 s temporal spectrum. A DMA of higher resolution would not only resolve perfectly the 3.4 and 2.9 particles (partially overlapping in Figure 8). Even more usefully, much narrower peaks would enable much faster scans without loss of resolution.

One could be led to conclude from the nucleation event illustrated in Figure 11 that a temporal resolution of 1 minute is adequate. This may be true for events lasting several minutes, but perhaps not in situations where the reservoir of vapor available for particle growth is more limited. Given that the improved time resolution of the instrument of Kong et al. enabled the capture of nucleation events that would have been missed with slower prior equipment, one may surmise that currently undetectable events will be capturable by future devices having more temporal resolution.

There is another potential advantage of higher resolution in nucleation studies, as well as in other situations with a natural mechanism to produce narrow size distributions. If the signal is concentrated into a narrow range of sizes, being able to resolve them would increase the signal. In Figure 8 of Kong et al. one sees that the peak corresponding to 3.4 nm particles spans a size range from 2.9 to about 3.9 nm. The atmospheric nucleation spectrum shown in Figure 11 includes mobility peaks of comparable width, suggesting that perhaps the real size distribution along the growth curve is considerably narrower and taller than can be captured with the available resolving power. In other words, higher resolution would enable detecting lower intensity as well as shorter nucleation events. It would also provide a more precise measurement of the growth rate.

Response:

Operating the mobility classifier at higher flow rates than those of the ROMIAC reported here would reduce the time required by reducing the residence time within the instrument.  The reviewer has demonstrated instrument scans as rapid as 1.2 s, using a "half-mini" DMA with a sheath flow rate of ~488 L/min, 40 times that used in the ROMIAC, and an aerosol flow rate of 3 L/min.  This enables the fast response that the reviewer describes.  For chamber experiments, the high sheath flow rate would require either operating the DMA with a recirculating sheath flow, or supplying the sheath flow from a different source than the sample to avoid depleting the aerosol from the chamber.  Either of these modes could perturb the aerosol, though the short residence time might make the effect of the thermodynamic perturbations small for some aerosols and measurement scenarios.  The reviewer correctly notes that improved size resolution may better reveal the fine structure of the particle size distribution.  Both of these potential benefits come at a cost of reduced  counts, and correspondingly increased statistical uncertainty.  Each measurement scenario involves compromises.  The ROMIAC allowed resolution comparable that employed in a wide range of atmospheric and simulated atmospheric measurements.  Time response and resolution are not the only considerations in atmospheric measurments.

The reviewer further notes that the data shown in Fig. 11 might benefit from both improvements identified.  That intense nucleation event produced large number concentrations, but many of the events that we studied produced lower number concentrations; ambient atmospheric concentrations are also often much smaller.  The number of particles detected is further reduced by the low charging probability for particles in the low nanometer size range, i.e., O(1%) for a bipolar diffusion charge conditioner as is commonly used in mobility size distribution measurements.  Furthermore, an experiment that operates nearly continuously (seven days per week, 24 hours per day) for more than a month does not lend itself to fine tuning of operating parameters that may be appropriate for laboratory experiments that can readily be repeated.  The finite chamber volume further constrains the sample flow rates that can be tolerated without depleting the air in the chamber.

3. Soft X-ray ionizer. This element does not receive as detailed a description as other components of the system. The choice of a bipolar source is defended based on the freedom it offers to examine positive and negative polarities. This is indeed a most useful feature, especially for particles of such small sizes, whose charging probability has not been yet well studied. Soft X-rays have an evident regulatory advantage over radioactive sources. However, all ionization sources relying on energetic particles pose a danger of converting organic volatiles into involatile species, causing artificial nucleation events in sufficiently polluted atmospheres. The details of how this tendency is moderated by Kong et al. would be of interest to the reader. In this realm, it is worth noting that the advent of electrospray ionization did change this situation long ago in the case of unipolar sources. One relevant feature of electrospray ionization at atmospheric pressure is that, in contrast to electrical discharges, ions are created under strictly thermal conditions. Vapors are certainly introduced through the evaporation of a solvent. However, in high conductivity electrosprays, solvent flow rates may be as small as $10^{-7}$ g/s. For this reason, electrosprays of pure volatile solvents containing volatile salts such as ammonium acetate have had an increased use in chemical analysis as a clean and efficient ionization source for vapors. More recently, bipolar electrospray sources combining a positive and a negative emitter have provided a clean substitute for bipolar sources based on ionizing radiation. There is some literature on the ionization probability of unipolar ES sources with vapors, and some untested calculations of how this probability would depend on particle size.  on the size dependence of the

charging probability of nanoparticles in bipolar ES sources, but a similar ambiguity exists for bipolar sources of ionizing radiation. Hopefully these two gaps will be filled soon.

Response:

As noted above, the unique component of this measurement system is the scanning ROMIAC, other charge conditioners (chargers), even other soft x-ray charge conditioners, or the ones that the reviewer proposes, could be used in a nSEMS.  Due to laboratory shutdowns during the COVID-19 pandemic, the characterization of the charge conditioner used in this study could not be completed in time to be included in this paper.  That characterization will be completed, and will be reported along with the design details in a separate paper.  We therefore propose to state in this paper indicate that we characterize that portion of the nSEMS downstream of the charge conditioner.  This level of characterization is needed since an important class of applications is its use as the analyzer in TDMA-type measurements (e.g., replacing the analyzer SMPS of a tandem differential mobility analyzer).

Though we are removing the charge conditioner employed from the instrument that we describe, the comments by the referee still warrant discussion.  Differential mobility analysis of environmental aerosols requires a known charge distribution if one is to determine the particle size distribution.  To that end, most DMA measurements employ a bipolar charge conditioner.  The reviewer suggests that choice of the bipolar charge conditioner is based upon the ease of selection of positive or negative ions.  While we take advantage of having both polarities,  the reason that bipolar charge conditioning is standard in mobility-based size distribution measurements of aerosols lies in the consistency of the charge distribution that results.  Exposing the aerosol to a cloud of ions of both polarities that is overall neutral leads to a steady-state charge distribution provided the concentration of particles is not so high that the ions are depleted during charge conditioning.  The neutral cloud of ions is typically produced by irradiating the sample with energetic particles, often those emitted by radioactive decay, but increasingly and, in this study, soft x-rays.   Detail on the soft x-ray source that we have employed is, admittedly, limited, since planned experimental characterization of it was delayed by the COVID-19 restrictions.  That work is beginning now, and will be the subject of a future paper.

The reviewer cautions that the energetic particles can lead to transformations of organic vapors that lead to particle formation, and false particle counts.  Whether we use a radioisotope or soft x-rays to generate the ions, we do detect "charger ions" in the 1-1.4 nm mobility equivalent diameter size range.  We have not resolved whether the charger ions might be particles as the reviewer suggests.  Since we cannot definitively discriminate these ions from the particles that we seek to measure, we only report size distributions for particles larger than 1.5 nm. Estimations of the nucleation rate in the CLOUD experiments and elsewhere are inferred from the particle flux through size space at a specific size (e.g., 1.7 nm) that is sufficiently large compared to the charger ions to allow confidence that those clusters or particles were present in the sampled air.

The reviewer further proposes using a pair of electrospray sources to produce ions of both polarities that would then be introduced into the aerosol to avoid new particle formation by energetic particles.  That might work, provided the electrospray sources are sufficiently robust to maintain charge balance and sufficient ion concentrations within the charge conditioner continuously for the duration of the experiment.  In the case of measurements at CLOUD or

atmospheric measurement campaigns, continuous operation for weeks or months is required, and the instrument must be able to operate unattended. The present methods of charge conditioning serve our purpose well, but do complicate measurements in the range where the charger ions are found. New approaches to charge conditioning that overcome this limitation would be welcome.

We agree with the reviewer that a broad discussion of the many design constraints that need to be considered in designing this or other types of instruments would be valuable, but this would require a comprehensive review of these many dimensions of instrument design in the context of different use scenarios. This paper describes an instrument that was designed for measurement of ultrafine particles in the atmosphere and in chamber studies that simulate the atmosphere, we limit our discussion to constraints that were considered for this specific instrument.

References cited:

Kuang, C, M Chen, PH McMurry Aerosol Science and, 2012. n.d. "Modification of Laminar Flow Ultrafine Condensation Particle Counters for the Enhanced Detection of 1 Nm Condensation Nuclei." *Taylor & Francis*.

Hering, Susanne V, Gregory S Lewis, Steven R Spielman, Arantzazu Eiguren-Fernandez, Nathan M Kreisberg, Chongai Kuang, and Michel Attoui. 2017. "Detection Near 1-Nm with a Laminar-Flow, Water-Based Condensation Particle Counter." *Aerosol Science and Technology* 51 (3): 354–62. doi:10.1080/02786826.2016.1262531.

---

## Author Comment (AC2)

Reviewer 2

*The nanoscaning electrical mobility spectrometer is a very useful instrument because of its response time and sensitivity for small particles.*

*Few questions or methods are not discussed in the text or are not clear. I listed few of them below. That will help the readers to have the responses. It will help to precise that the particles used to calibrate the instrument are electrically (positive) charged. Sizing sub 2 nm with a charger dma cpc is another problem to my opinion. Indeed the chemistry of the particles is changed by the attached ion on the particle. And the chemistry is important for the activation of sub 2 nm particles, see Kangasluoma et al (2013) in Aerosol Sci Technol.;*

We appreciate the reviewer's pointing out this very insightful paper. Kangasluoma et al. (2013) observed that composition more strongly affects detection efficiency than does charge state for CPC detection of mobility classified molecular ions smaller than about 1.5 nm. Furthermore, in comparing DMA/CPC signals to high resolution mass spectrometric analysis of the classified inorganic particles, they found that a substantial fraction of particles at 1.5 nm did not match the composition of the generated particles, making the concentrations of those very small particles uncertain. This key reference supports our inference, and suggests reasons that the size distribution below about 1.5 nm is difficult to quantify with the present instrument. We will cite this paper in the revised manuscript.

Regarding the charge state of the aerosol, and its impact on the CPC performance: we note that the focus of this work is on the nanoSEMS system. The CPC testing reported in this manuscript only addresses use of the CPC as a detector for a mobility analysis system. As such, the aerosol used in testing the CPC has all been performed with mobility classified, and, therefore, charged particles. The CPC counting efficiency data presented in Fig. 6 are for positively charged particles.

*Jiang et al. 2011 in the same journal have used by the way the concept of SMPS to detect neutral particles in the sub 2 nm range. You should cite their work.*

Jiang et al. (2011a) should indeed be cited for the first development of the DEG-SMPS that extended the sizing range of an SMPS based on the TSI nano-DMA to sizes approaching 1 nm, as well as for the first application of the instrument in concert with mass spectrometric methods that show that the two methods yield consistent size distributions in the size regime where they overlap (Jiang et al., 2011b). We will cite these important works.

*Regarding enumerated comments:*

We note that the primary focus of this paper is the scanning-mode ROMIAC, which has enabled creation of the nSEMS. The integrated nSEMS system also requires a charge conditioner (charger), and a detector (a condensation particle counter, CPC). Other charge conditioners, even others based upon soft x-ray sources, could be substituted for the present design without changing the essential characteristics of the instrument. Similarly, other detectors than the two-stage condensation particle counter (CPC) that we developed

and used as the detector within the nSEMS could be subsituted in the nSEMS, altering its performance in detail, but not its essential character.

Due to delays caused by the COVID-19 pandemic, full characterization of the soft x-ray charge conditioner employed in this study could not be completed in time for this study. The necessary experiments will soon resume; the design of the charge conditioner and its performance (particle losses and output charge distribution) will be the subject of a separate paper. The only place where the charge conditioner affects results presented in this paper is example of its application presented in Fig. 11. The assumptions made regarding the performance of the charge conditioner are discussed in the responses to queries given below, and will be made clear in the revised manuscript.

With that caveate, this paper focuses on the integrated measurement system, and provides the detail needed to analyze the data that it generates. We provide greater detail on the core component of the nSEMS, i.e., the classifier (ROMIAC), than we do on the soft x-ray charge conditioner or the two-stage CPC. However, as discussed below in response to specific questions, the performance of the two-stage CPC is taken into account in the calibration studies through which we determined the transfer function of the *integrated instrument* beginning downstream from the charge conditioner. We feel that the information provided is sufficient to understand, or quantitatively interpret the data from the *nanoSEMS system*.

Work is continuing that will provide the level of detail requested in some of the reviewer's enumerated comments, and will be presented in future papers. Given that data obtained with the nSEMS have already formed the basis of published papers, we would like to avoid the additional delays that the additional work on the subcomponents will require.

*1°/ Why this 'precise' diameter of 1.5 nm? Is it a limit of the detector or the classifier or the charger. Tetraethylammonium bromide produces 1.11 nm (mobility diameter) has been used by Attoui (2018 Journal of aerosol science).*

In the original work on the mobility classifier (the ROMIAC) used in the nSEMS, Mui et al. (AST, 2017) calibrated the ROMIAC in voltage stepping mode using tetra-alkylammonium halides, including tetraethylammonium bromide. The ROMIAC can extend the sizing range to 1.1 nm size. While it would be great to report size distributions closer to 1 nm, in ambient or atmospheric simulation chamber measurements, counts are recorded for sub-1.5 nm particles, even when none are present. In measurements of inorganic calibration particles, Kangasluoma et al. (2013) observed many impurity counts in this size range that were attributed to impurities. Hence, we limit the reported sizing range to that in which we have confidence that we are measuring particles from our sample rather than impurities or gas ions. This will be clarified in the revised manuscript, and the Kangasluoma paper will be cited.

*2°/ The charger is the weak point of the study since there is no experimental work on the charging state of sub 3 nm with a bipolar charger. The authors are not giving lot of details about the charger they have used in terms of size of their chamber, concentration of ions nor residence time of the ions in the chamber.*

As noted above, we will revise the paper to make it clear that this paper addresses the integration of the scanning-voltage ROMIAC with the two-stage CPC to create the nSEMS, and that the soft x-ray charge conditioner has not been fully characterized at this time.

The charging probability is a major weakness in all mobility-based measurements of ultrafine particles. The soft x-ray charge condition is a new design developed for this instrument, but which has been used on numerous other measurements. It was designed to minimize losses by avoiding the tortuous paths found in other soft x-ray chargers, and by emplying a short residence time (which also lowers the risk of new particle formation within the irradiated volume (Yun et al., 2009)). In initial characterization experiments, it yielded consistent results with conventional radio-isotope ($^{210}$Po and $^{85}$Kr) charge conditioners in experiments in both chamber and calibration experiments, and applying the Wiedensohler (1988) empirical fit to the theoretically-derived, steady-state, bipolar charge distribution (Fuchs, 1963; Hoppel and Frick, 1986), as is commonly applied to DMA data analysis.

Figure 4 and the associated text will be modified to indicate that the space occupied by the charge conditioner was replaced with a straight tube of the length of the charge conditioner during the calibration experiments. The concentration of the classified particles entering the nSEMS was measured either with an aerosol electrometer, or a reference CPC. The response of the nSEMS, including the two-stage CPC is then measured.

*3°/The important flowrate of 4.6 lpm is coming without any explanation about the charging state nor residence time.*

*4°/ Same thing about the losses versus the particles size in the charger, in the conditioner after the dma?*

The input flow rate to the charge conditioner was selected based upon prior experience sampling from the CLOUD chamber for measurements using other instruments, combined with design simulations; the charge conditioner design was optimized for that flow rate. Losses in the charge conditioner do not affect the calibration data. They may have a modest effect the size distributions shown in Fig. 11 for the one application example that we report for the instrument; a caveate to this effect will be added to the discussion of those data.

*5°/ Likewise about the mixing chamber of the mixing type cpc activator. There are no details on terms of size nor activation nor losses.*

*6°/ Likewise about the growth tube (if there is a growth tube) of the mixing type CPC. In page 7 line 47, the authors are talking about the growth tube used by Sgro and Fernandez de la Mora indeed. There are no details about the residence time the inner diameter nor length. Is the growth tube what they call 'condenser'? What is its length and inner diameter?*

*7°/ Same thing for the flow rate of '1, 5 lpm'. Why this particular value?*

As noted above, the focus of this paper is on the nSEMS, which incorporates a two-stage CPC, but the unique component is the scanning-mode radial opposed migration aerosol classifier. The mixing type CPC is based upon the fast-mixing CPC design of Wang et al.

(2002), as modified by Shah and Cocker (2006), which affords a shorter response time than that of the particle size magnifier of Vanhanen et al. (2011).   It was originally designed to supply activated particles to a TSI Model 3760 CPC butanol CPC operating at a flow rate of 1.5 L/min.  That detection stage was replaced with a high flow (1 L/min) version of the Aerosol Dynamics Inc. water-based MAGIC CPC, operating at 1 L/min, improving the nSEMS time response due to the ADI detector's faster response.  The 1.5 L/min flow was maintained; the excess flow was exhausted upstream of the MAGIC CPC.

Two aspects of the CPC performance are central to the interpretation of data from the nSEMS: (i) the counting efficiency as a function of size, and (ii) the time response of the detector system.  The counting efficiency of the CPC was separately determined using the apparatus shown in Fig. 5, and  is presented in Fig. 6.  The integrated nSEMS transfer function was measured for a very slow scan (1400 s) for a number of sizes of mobility-classified particles, providing data that closely approximate the constant voltage transfer function.   It was also measured at the scan rate used in the experiments (60s).

By deconvoluting the two transfer function measurements, we extracted the time response of the integrated detector plus the ROMIAC and downstream plumbing. That time response was fitted to a model incorporating a single, well-mixed volume (a CSTR  with an exponential decay in delay times) in series with a fixed delay (often described as a plug-flow reactor.  The resulting response time function, which has an exponential decay time of 0.2 s, and a fixed delay of 0.7s, is shown in Figure S2.  This characterization was employed in analysis of the data since the response time of the first (turbulent mixing) stage of the CPC was not separately measured.  The time response of the second-stage detector CPC was sigificantly slower than the estimated exponential decay time of the mixing region of the first stage, 7.5 ms, which is slightly shorter than that measured by Shah and Cocker (2005) for an earlier version of the first stage (10 ms).  The full residence time of the first stage is ~0.12 s, including the mixing region and the growth tube (which was labeled condenser in the original manuscript but will be changed to match the terminology that has become common in the literature).   Delays caused by mixing and by the finite volume within the CPC and elsewhere shift the peak in the transfer function.  Figure 10 shows that, after accounting for those delays in data inversion for a 60 s scan, measurements agree well with that predicted in COMSOL simulations of the scanning ROMIAC.

Thus,  while we do not present full characterization of the mixing particle activation stage or separate measurements of the transient response of the two-stage CPC, the data needed to interpret and invert the data were measured and are reported.

*8°/ A hot wire is used for the calibration of the instrument but there are no information nor reference of this method. Is it the same method used by Peineke et al 2009 (Journal of aerosol science)? If yes why the authors are using a charger in the figure 5. Peineke and Schmidt Ott 2007 (Journal of Aerosol Science) claim that the particles are self-charged in negative and in positive mode.*

The device that we used is similar to that of Peineke et al. (2006).  We will cite that paper.

*9°/ Why the charger (neutralizer) is not used any more in the figure 5 for the same hot wire generator?*

As reported by Peineke et al. (2006), many of the particles generated by the hot-wire source are positively charged when formed. Passing the very small particles through a bipolar charge conditioner reduces the number of charged particles. A comment to this effect can be added to the paper.

*10°/ The PSM as a sizer has been introduced by Gamero and Fermandez de la Mora (paragraph 3.2 in Gamero 2000 Journal of aerosol science). Not by Sgro and Fernandez de la Mora (2004) as said the authors.* **Citation**: *https://doi.org/10.5194/amt-2021-62-RC2*

We thank the reviewer for pointing out this error. The proper paper will be cited.

References:

Fuchs, N A. 1963. "On the Stationary Charge Distribution on Aerosol Particles in a Bipolar Ionic Atmosphere." *Geofisica Pura E Applicata* 56 (1): 185–93. doi:10.1007/BF01993343.Hoppel and Frick (1986)

Jiang, Jingkun, Jun Zhao, Modi Chen, Fred L Eisele, Jacob Scheckman, Brent J Williams, Chongai Kuang, and Peter H McMurry. 2011. "First Measurements of Neutral Atmospheric Cluster and 1–2 Nm Particle Number Size Distributions During Nucleation Events" 45 (4): 2–5.

Jiang, Jingkun, Modi Chen, Chongai Kuang, Michel Attoui, and Peter H McMurry. 2011. "Electrical Mobility Spectrometer Using a Diethylene Glycol Condensation Particle Counter for Measurement of Aerosol Size Distributions Down to 1 Nm." Aerosol Science and Technology 45 (4): 510–21. doi:10.1080/02786826.2010.547538.

Kangasluoma, J, H Junninen, K Lehtipalo, J Mikkila, J Vanhanen, M Attoui, M Sipila, D Worsnop, M Kulmala, and T Petaja. 2013. "Remarks on Ion Generation for CPC Detection Efficiency Studies in Sub-3-Nm Size Range." *Aerosol Science and Technology* 47 (5): 556–63. doi:10.1080/02786826.2013.773393.

Peineke, C, M B Attoui, and A Schmidt-Ott. 2006. "Using a Glowing Wire Generator for Production of Charged, Uniformly Sized Nanoparticles at High Concentrations." *Journal of Aerosol Science* 37 (12): 1651–61. doi:10.1016/j.jaerosci.2006.06.006.

Shah, SD, and DR Cocker. 2005. "A Fast Scanning Mobility Particle Spectrometer for Monitoring Transient Particle Size Distributions." *Aerosol Science and Technology* 39 (6): 519–26. doi:10.1080/027868291004652.Vanhanen (2011)

Wang, J, VF McNeill, DR Collins, and RC Flagan. 2002. "Fast Mixing Condensation Nucleus Counter: Application to Rapid Scanning Differential Mobility Analyzer Measurements." *Aerosol Science and Technology* 36 (6): 678–89. doi:10.1080/02786820290038366.

Wang, J, VF McNeill, DR Collins, and RC Flagan. 2002. "Fast Mixing Condensation Nucleus Counter: Application to Rapid Scanning Differential Mobility Analyzer

Measurements." *Aerosol Science and Technology* 36 (6): 678–89. doi:10.1080/02786820290038366.

Yun, Ki Myoung, Sin Young Lee, Ferry Iskandar, Kikuo Okuyama, and Naoko Tajima. 2009. "Effect of X-Ray Energy and Ionization Time on the Charging Performance and Nanoparticle Formation of a Soft X-Ray Photoionization Charger." *Advanced Powder Technology* 20 (6): 529–36. doi:10.1016/j.apt.2009.07.002.

---

## Author Response (AR1)

Revisions summary

The nano-scanning electrical mobility spectrometer (nSEMS)
and its application to size distribution measurements of 1.5-25 nm particles

5 **Revisions in response to Reviewer 1**

Reviewer 1 identifies a number of interesting questions, but they are generally beyond the scope of this paper, pointing to alternative ways to assemble an instrument. The reviewer explicitly states:

> They have accordingly no duty to engage in the more extensive discussion I propose. Nevertheless, given their wide experience, their response would be exceedlingly useful to the many colleagues involved in these measurements,
> 10 as well as on the many possible alternative experimental tools that were not chosen.

We have provided extensive responses to the reviewer's comments in the on-line response. A thorough analysis of each of the points would significantly lengthen the paper, and would detract from the objective of the paper which is to describe an instrument that has already yielded important insights into new particle formation under simulated atmospheric conditions. This may be a useful topic for a review or tutorial paper, one that addresses the philosophy of measurement design. The is 15 always a danger of allowing the desire for perfection to prevent getting the data that is needed to answer the scientific question that one is trying to address. Compromise is often necessary in the interest in having an instrument that will reliably provide the required data when the instrument must be deployed.

We, therefore, decline to add the additional dimensions to this paper that the reviewer suggests, but will certainly consider ways to open the discussion of the important questions that he raises in the literature in the near future.

20

**Revisions in response to Reviewer 2**

*The nanoscaning electrical mobility spectrometer is a very useful instrument because of its response time and sensitivity for small particles.*

25 *Few questions or methods are not discussed in the text or are not clear. I listed few of them below. That will help the readers to have the responses. It will help to precise that the particles used to calibrate the instrument are electrically (positive) charged. Sizing sub 2 nm whith a charger dma cpc is another problem to my opinion. Indeed the chemistry of the particles is changed by the attached ion on the particle. And the chemistry is important for the activation of sub 2 nm particles, see Kangasluoma et al (2013) in Aerosol Sci Technol.; Jiang et al. 2011 in the same journal have used by the way the concept of SMPS to detect* 30 *neutral particles in the sub 2 nm range. You should cite their work.*

Reference to Kangasluoma et al. (2013) has been added with discussions as to how the results from Kangasluoma show that more counts are recorded for "particles" smaller than 1.5 nm than are present in the incoming sample (as measured by mass spectrometer), leading to considerable ambiguity and uncertainty in size distributions below this size. Hence we do not claim to measure size distributions below 1.5 nm. The following text has been added:

35 > While this CPC can detect particles as small as 1 nm diameter, and the ROMIAC can classify particles of that small
> size (Mui et al., 2013, 2017), Kangasluoma et al. (2013) observed larger numbers of apparent particles smaller than
> 1.5 nm diameter than were present in the calibration aerosol that they generated, and attributed the excess particle
> counts to the activation of gas ions within the CPC. Lacking a definitive method for discriminating between gas
> ions and particles in the sub-1.5 nm size range, we do not report size distributions below 1.5 nm.

40 *Jiang et al. 2011 in the same journal have used by the way the concept of SMPS to detect neutral particles in the sub 2 nm range. You should cite their work.*

We have added at line 67 in the annotated version of the document:

> ..., which was integrated with a DMA to produce the first SMPS system to measure size distributions approaching
> 1 nm (Jiang et al., 2011a) that agree closely with mass spectrometric data in the overlap region between the two
> 45 methods (Jiang et al., 2011b).

*1°/ Why this 'precise' diameter of 1.5 nm? Is it a limit of the detector or the classifier or the charger. Tetraethylammonium bromide produces 1.11 nm (mobility diameter) has been used by Attoui (2018 Journal of aerosol science).*

As noted above, Kangasluoma et al. (2013) performed calibration experiments using particles of known composition in which they observed more counts than there were calibration particles in the aerosol sample flow. In our experiments, we observed such phantom counts as well, interpreted as gas ions. We have cited the Kangasluoma work to support our decision not to claim that we have robust particle size distribution measurements below 1.5 nm where these phantom counts make the measurements ambiguous and uncertain. This ambiguity remains an important challenge for aerosol measurements.

*2°/ The charger is the weak point of the study since there is no experimental work on the charging state of sub 3 nm with a bipolar charger. The authors are not giving lot of details about the charger they have used in terms of size of their chamber, concentration of ions nor residence time of the ions in the chamber.*

We agree that the lack of characterization of the charge leads to an incomplete description of the full measurement system employed in the the measurements shown in Fig. 11. The emphasis of this paper is on the use of the opposed migration classifier in voltage-scanning mode to determine particle size distributions in the low nanometer size range. All of the instrument characterization experiments and data begin with a charged aerosol. The reference concentrations for all of those measurements are the concentrations of particles that can be classified and detected, i.e., mobility-classified particles that were charged either by the soft x-ray charge conditioner, or were produced as charged particles from the hot-wire source. The instrument-characterization/calibration data are, therefore, not compromised in any way by the lack of calibration data on the charge conditioner. The charge conditioner has shown no clear differences from other radioisotope or soft x-ray charge conditioners in preliminary tests, but completion of those tests was delayed by laboratory closure due to COVID-19. The charge distribution is consistent with the Wiedensohler (1988) fit to the Hoppel and Frick (1988) calculations using Fuchs (1964) theory. We have used that distribution in the analysis of data for Figure 11, which is the only place where the charge distribution enters the analysis in this paper.

The charge-state of the aerosol is the biggest source of uncertainty in mobility measurements of particle size distributions in general, even for conventional DMAs. We are, therefore, undertaking detailed characterization of this charge conditioner, as well as others to which we have access. This will be the subject of a future paper.

Rather than dwelling on the causes for delay, we have redefined the nSEMS as consisting of the ROMIAC classifier and the two-stage CPC, and pointed out that the soft x-ray source provides the required charged aerosol for the calibration studies reported in this paper. This has required a number of edits throughout the paper and the abstract. Key revisions include:

The Abstract now states:

The nSEMS consists of a novel differential mobility analyzer, and a two-stage condensation particle counter (CPC). The mobility analyzer, a radial opposed migration ion and aerosol classifier (ROMIAC), can classify nanometer-sized particles with minimal degradation of its resolution or diffusional losses.

The soft x-ray charge conditioner is mentioned at the end of the Abstract as:

Using a soft x-ray bipolar ion source in a compact housing designed to optimize both nanoparticle charging and transmission efficiency as a charge conditioner, the nSEMS has enabled measurement of the contributions of both neutral and ion-mediated nucleation to new particle formation.

The final paragraph of the introduction now reads:

In this work we show the development of a nano-scanning electrical mobility spectrometer (nSEMS) that features a fast-scanning OMAC, and a two-stage CPC, to acquire fast and accurate particle size distributions in the range of 1.5 - 25 nm. Here we describe the design and characterization of the scanning OMAC in detail as that is the unique component of the instrument which enables a sizing range that is not possible with a constant-flow DMA.

A radial-flow version of the OMAC that is capable of classifying charged particles or ions as small as 1 nm in diameter (Mui et al., 2013, 2017) is used in this instrument. The two-stage CPC includes a fast-mixing activation stage using DEG as working fluid, followed by an eco-friendly, fast-response, water-based CPC (Hering et al., 2019). Other CPCs, either two-stage ones like the one we have employed, or single stage CPCs that are capable of counting particles as small as 1 nm could be integrated into the nSEMS, so we limit our discussion of the 2-stage CPC to its essential features, its integration into the nSEMS, and the resulting performance. The nSEMS analyzes a charged aerosol, which can be naturally charged or one that is processed through a charge conditioner (CC). The present paper focuses on nanoparticle sizing and detection downstream of the CC. The integrated transfer function of the nSEMS system was derived based on both experimental results and finite-element modeling using COMSOL Multiphysics$^{TM}$. The nSEMS has been intensively used in the Cosmics Leaving OUtdoor Droplets (CLOUD) experiments at CERN, in which its size resolution and fast response have made it possible to follow very rapid growth of freshly nucleated nanoparticles, and to identify a new mechanism for new particle formation in a highly polluted atmosphere (Wang et al., 2020). A comparison of nSEMS data with measurements from other well-calibrated particle sizing instruments at CLOUD confirms its capacity to provide reliable size distribution in the low-nanometer size regime.

In the discussion of the nSEMS system, we now state:

The aerosol enters the nSEMS after passing through a soft x-ray CC at a relatively high flow rate of 4.6 L min$^{-1}$. A smaller, 1.2 L min$^{-1}$ polydisperse aerosol sample flow, from the core of the larger flow is introduced into a ROMIAC, while the remainder is exhausted. The high flow rate through the CC and the core-sampling flow spitter are designed to minimize losses of the highly diffusive nanoparticles. While the ROMIAC was originally designed to operate at constant voltage, the voltage is continuously varied in an exponential ramp in the nSEMS. Particles transmitted through the ROMIAC are counted using a two-stage CPC to capture the particle size distribution. Detailed operating parameters and default settings are summarized in Table 1.

The CC used in this study employs a soft x-ray source to produce ion pairs from the gas molecules in order to bring the aerosol to a steady-state Fuchs charge distribution that enables estimation of the size distribution whose initial charge state is uncertain. The soft x-ray source has two advantages: (*i*) fewer regulatory hurdles than radio-isotope sources used for the same purpose, and (*ii*) the ability to turn off the ion source in order to measure those particles in the sample that carry charge due to gas ionization by galactic or simulated cosmic rays. It was designed to minimize losses of the smallest particles. Preliminary characterization of the CC, and data from numerous experiments in which it has been applied on conventional DMAs in parallel with other DMAs using conventional CCs, and on the nano-radial differential mobility analyzer (Brunelli et al., 2009) yield results consistent with the Wiedensohler (1988) approximation to the Fuchs charge distribution, with no apparent biases. Since mobility classified, and, therefore, charged particles are used to determine the integrated instrument transfer function that is reported here, the key results from this paper are not affected by any minor deviations from the Fuchs charge distribution that is assumed in inverting data from the nSEMS. Details of the design and quantitative calibration of the soft x-ray CC will be presented in a separate paper.

Wording has been changed slightly in a number of other places, including the conclusions, to keep the message consistent with excluding the soft x-ray charge conditioner from the bounds of the nSEMS as we now define it.

*3°/The important flowrate of 4.6 lpm is coming without any explanation about the charging state nor residence time.*

The 4.6 L/min flow is designed to minimize losses in the soft x-ray charge conditioner. This will be discussed in the detailed paper on that device.

*4°/ Same thing about the losses versus the particles size in the charger, in the conditioner after the dma?*

These losses do not affect the calibration studies that are the focus of this paper. They could affect the data presented in Fig. 11, but the high concentration seen at the smallest reported sizes suggest that the losses are not excessive, as do design simulations that will be part of the paper describing the charge conditioner. The losses in the soft x-ray charge conditioner will be quantified in the separate paper that we will write on that device.

*5°/ Likewise about the mixing chamber of the mixing type cpc activator. There are no details on terms of size nor activation nor losses.*

The discussion of the turbulent mixing activation chamber within the two-stage CPC has been enhanced somewhat, and now reads:

Classified particles transmitted through the ROMIAC are subsequently detected by a two-stage CPC that enables particle counting approaching 1 nm in size (Iida et al., 2009; Jiang et al., 2011a). The first stage employs a fast-mixing condensational activation and growth reactor (Wang et al., 2002) that uses DEG as the working fluid to activate the nanoparticles. This activation stage is based upon the particle size magnifier (PSM) (Kousaka et al., 1982; Okuyama et al., 1984; Gamero-Castano and de la Mora, 2000; Sgro and de la Mora, 2004) in which a cool aerosol sample flow undergoes rapid turbulent mixing with a warm flow that is saturated with vapor to produce the supersaturated state that will activate and grow particles larger than a critical size. The detailed design incorporates modifications reported by Shah and Cocker (2005) that reduce the size of the mixing volume, while maintaining the small growth tube, with a residence time of $\sim 10$ ms. In the first stage used in this study, supersaturation is produced by turbulent mixing of a 0.3 L min$^{-1}$ flow of hot (e.g., 70 °C) DEG vapor with a 1.2 L min$^{-1}$ cold (20 °C) particle-laden flow. The downstream growth tube is cooled to 10 °C to accelerate particle growth and remove excess vapor. In contrast to the Airmodus particle size magnifier (PSM) and the CPC of Sgro and de la Mora (2004), on which the PSM is based, the mixing time in the activation stage of the present CPC has been minimized ($\sim 0.12$ s) to speed instrument response, as compared with $\sim 0.7$ s. The design and initial experiments were performed a TSI Model 8210 CPC operating at a flow rate of 1.5 L min$^{-1}$ as the second detection stage. Owing to the slow response of that CPC, it was replaced with a modified, water-based ADI MAGIC$^{TM}$ CPC that serves as the second stage to grow particles sufficiently large for optical detection (Hering et al., 2019). Particle counts are recorded over the nSEMS size distribution scan at 5 Hz. The sample flow rate of the CPC is 1.00 L min$^{-1}$. Between the activation and booster stages, the flow is split between the water-CPC and a smaller (0.5 L min$^{-1}$) excess flow to minimize deposition of excessive DEG vapor in the intervening plumbing, and to match the flow to the water CPC.

The specific design follows from Wang et al. (2002), with modifications that we first proposed, but that were first implemented and reported by Shah and Cocker (2005). The residence time in the growth region (condenser) is now specified as requested.

*6°/ Likewise about the growth tube (if there is a growth tube) of the mixing type CPC. In page 7 line 47, the authors are talking about the growth tube used by Sgro and Fernandez de la Mora indeed. There are no details about the residence time the inner diameter nor length. Is the growth tube what they call 'condenser'? What is its length and inner diameter?*

What was originally labeled the condenser is now identified as the growth tube; it has a residence time of 120 ms, which is the relevant quantity for assessing its effect on data analysis. In this paper, we look at the two-stage CPC as a unit, rather than characterizing each sub-component. The larger second stage CPC dominates the time response. The counting efficiency of the entire two-stage CPC is reported in Fig. 6.

*7°/ Same thing for the flow rate of '1, 5 lpm'. Why this particular value?*

The 1.5 L/min flow rate was a design decision based upon components we had available. We initially used a butanol CPC as the second stage, but replaced it with a more reliable and faster response water CPC, which is the instrument that we report here.

*8°/ A hot wire is used for the calibration of the instrument but there are no information nor reference of this method. Is it the same method used by Peineke et al 2009 (Journal of aerosol science)? If yes why the authors are using a charger in the figure 5. Peineke and Schmidt Ott 2007 (Journal of Aerosol Science) claim that the particles are self-charged in negative and in positive mode.*

Hot wire sources for ultrafine partices have been used since before DMAs became available commercially. The device that we used in this study is similar to that of Peineke et al. (2006), so we have cited that work.

*9°/ Why the charger (neutralizer) is not used any more in the figure 5 for the same hot wire generator?*

The hot wire source generates an abundance of charged particles, so we used them directly.

*10°/ The PSM as a sizer has been introduced by Gamero and Fermandez de la Mora (paragraph 3.2 in Gamero 2000 Journal of aerosol science). Not by Sgro and Fernandez de la Mora (2004) as said the authors. Citation: https://doi.org/10.5194/amt-2021-62-RC2*

We have included the Gamero-Castano reference as requested, but for consistency have given proper credit to Okuyama and collaborators who introduced the method and the name of particle size magnifier. (There is an 1960 paper written in Russian, by Kogan; the reference that I have for it seems to be incorrect so I have not cited it.) The citation was for the use of the PSM to detect particles, not to size them, but this is a minor point, and not worth quibling over. We have kept the Sgro reference since her much simpler design is the one that is used in the Aermodus Particle Size Magnifier, and that determines the response time of that instrument as discussed in this paper.

**Other revisions:**

In the course of revising this paper, we found that the discussion of the data inversion method could be confusing, so we revised it to make the method that we used clearer. The revised text is:

Data inversion is required to retrieve particle size distribution of the source particles, $n(\log d_p)$, from the particle counts measured by the CPC, $\mathbf{R}_{\text{nSEMS}}$, which can be represented in matrix form as

$$\mathbf{R}_{\text{nSEMS}} = \mathbf{A}_{\text{nSEMS}}\mathbf{N}, \tag{1}$$

where $\mathbf{A}_{\text{nSEMS}}$ is often referred to as the inversion kernel for the instrument, and $\mathbf{N}$ is the vector of weights for the discretized representation of the particle size distribution, for which we use linear splines on $x = \log(d_p)$ (Mai et al., 2018). The time-series instrument response can be written as $\mathbf{R}_{\text{nSEMS}} = [R_{\text{nSEMS},1}, R_{\text{nSEMS},2}, \cdots, R_{\text{nSEMS},I}]^{\text{T}}$. With the default nSEMS voltage ramp time, $t_{\text{ramp}} = 50$ s, and the CPC data recording frequency, $t_{\text{c}} = 0.2$ s, the vector length for one complete scan is $I = 250$. The particle number counts recorded by the CPC in the $i^{\text{th}}$ time bin, $R_{\text{nSEMS},i}$, can be represented as the integral of the total number of particles transmitted over the time interval

$(i-1)t_\text{c} \le t < it_\text{c}$:

$$R_{\text{nSEMS},i} = Q_\text{a} \int\limits_{(i-1)t_\text{c}}^{it_\text{c}} \int\limits_{-\infty}^{\infty} n(x) \sum_{\phi} p_\text{charge}(x,\phi)\, \eta_\text{CPC}(x,\phi)\Omega_\text{nSEMS}(Z_\text{p}(x,\phi),\beta,\delta,t)\, \mathrm{d}x\, \mathrm{d}t$$

$$= Q_\text{a} \int\limits_{(i-1)t_\text{c}}^{it_\text{c}} \sum_{j} \int\limits_{x_{j-1}}^{x_j} n(x) \sum_{\phi} p_\text{charge}(x,\phi)\, \eta_\text{CPC}(x,\phi)\Omega_\text{nSEMS}(Z_\text{p}(x,\phi),\beta,\delta,t)\, \mathrm{d}x\, \mathrm{d}t \tag{2}$$

The particle charging probability from the soft x-ray CC, $p_\text{charge}(x,\phi)$, was assumed to be that of the Wiedensohler (1988) approximation, and is computed separately for scans of negative and positive polarity. In order to obtain the instrument transfer function, $\Omega_\text{nSEMS}(Z_\text{p}(x,\phi),\beta,\delta,t)$, for each time bin, the simulated ROMIAC transfer functions were first fitted as a Gaussian function:

$$\Omega_\text{ROMIAC}(Z_\text{p}(x,\phi),\beta,\delta,t) = a\exp\left(-\frac{(t-b)^2}{2c^2}\right) \tag{3}$$

The three fitting parameters, $a$, $b$, and $c$, were then interpolated over the entire time vector with 250 bins. By substituting the interpolated parameters back into Eq.(3), a ROMIAC transfer function, $\Omega_\text{ROMIAC}(Z_p(x,\phi),\beta,\delta,t)$, was generated for each time bin. The fitted transfer functions were adjusted by the empirically determined mobility correction factor, $f_z$. The nSEMS transfer function for each time bin, $\Omega_\text{nSEMS}(Z_\text{p}(x,\phi),\beta,\delta,t)$, was then computed by the convolution of the ROMIAC transfer function and the CPC residence time distribution of Eq.12. The inversion kernel matrix for the $i^\text{th}$ time bin, and $j^\text{th}$ particle size bin thus becomes

$$A_{nSEMS,i,j} = Q_a t_c \int\limits_{\log d_{p,j-1}}^{\log d_{p,j}} n(x) \sum_{\phi} p_\text{charge}(x,\phi)\, \eta_\text{CPC}(x,\phi)\Omega_\text{nSEMS}(Z_\text{p}(x,\phi),\beta,\delta,t)\, \mathrm{d}x \tag{4}$$

We then applied a totally nonnegative least squares (TNNLS) algorithm to retrieve the sample particle size distribution from the inversion kernel and the particle number concentrations detected by the CPC, i.e., solving for $N = \mathbf{A}_\text{nSEMS}^{-1}\mathbf{R}_\text{nSEMS}$ (Merritt and Zhang, 2005; Mai et al., 2018).

Other small revisions have been made to ensure consistency (especially with respect to the treatment of the charge conditioner, correct typographical errors, formatting of units, supply missing units Fig. 6, change the label of the growth tube on Fig. 1, and other details. All revisions (except those in the two figures), are identified in the PDF version of shows the differences between the present manuscript and the one that was originally submitted.

**References**

Brunelli, N. A., Flagan, R. C., and Giapis, K. P.: Radial Differential Mobility Analyzer for One Nanometer Particle Classification, Aerosol Science and Technology, 43, 53–59, https://doi.org/10.1080/02786820802464302, 2009.

Gamero-Castano, M. and de la Mora, J.: A condensation nucleus counter (CNC) sensitive to singly charged sub-nanometer particles, Journal of Aerosol Science, 31, 757–772, https://doi.org/10.1016/S0021-8502(99)00555-8, 2000.

Hering, S. V., Lewis, G. S., Spielman, S. R., and Eiguren-Fernandez, A.: A MAGIC concept for self-sustained, water-based, ultrafine particle counting, Aerosol Science and Technology, 53, 63–72, https://doi.org/10.1080/02786826.2018.1538549, 2019.

Iida, K., Stolzenburg, M. R., and McMurry, P. H.: Effect of Working Fluid on Sub-2 nm Particle Detection with a Laminar Flow Ultrafine Condensation Particle Counter, Aerosol Science and Technology, 43, 81–96, https://doi.org/10.1080/02786820802488194, 2009.

Jiang, J., Chen, M., Kuang, C., and Attoui, M.: Electrical mobility spectrometer using diethylene glycol condensation particle counter for measuring aeerosol size distributions down to 1nm, Aerosol Science and Technology, 45, 510–521, https://doi.org/10.1080/02786826.2010.54738, 2011a.

Jiang, J., Chen, M., Eisele, F. L., Scheckman, J., Williams, B. J., Kuang, C., and McMurry, P. H.: First measurements of neutral atmospheric cluster and 1 − 2 nm particle number size distributions during nucleation events, Aerosol Science and Technology, 45, 2 − 5, https://doi.org/10.1080/02786826.2010.546817, 2011b.

Kangasluoma, J., Junninen, H., Lehtipalo, K., Mikkilä, J., Vanhanen, J., Attoui, M., Sipilä, M., Worsnop, D., Kulmala, M., and Petäjä, T.: Remarks on Ion Generation for CPC Detection Efficiency Studies in the Sub-1-nm Size Range, Aerosol Science and Technology, 47, 556–563, https://doi.org/10.1080/02786826.2013.773393, 2013.

Kousaka, Y., Niida, T., Okuyama, K., and Tanaka, H.: Development of a mixing type condensation nucleus counter, Journal of Aerosol Science, 13, 231–240, https://doi.org/10.1016/0021-8502(82)90064-7, 1982.

Mai, H., Kong, W., Seinfeld, J. H., and Flagan, R. C.: Scanning DMA data analysis II. Integrated DMA-CPC instrument response and data inversion, Aerosol Science and Technology, 52, 1400–1414, https://doi.org/10.1080/02786826.2018.1528006, 2018.

Merritt, M. and Zhang, Y.: Interior-Point Gradient Method for Large-Scale Totally Nonnegative Least Squares Problems, Journal of Optimization Theory and Applications, 126, 191–202, https://doi.org/10.1007/s10957-005-2668-z, 2005.

Mui, W., Thomas, D. A., Downard, A. J., Beauchamp, J. L., Seinfeld, J. H., and Flagan, R. C.: Ion Mobility-Mass Spectrometry with a Radial Opposed Migration Ion and Aerosol Classifier (ROMIAC), Analytical Chemistry, 85, 6319–6326, https://doi.org/10.1021/ac400580u, 2013.

Mui, W., Mai, H., Downard, A. J., Seinfeld, J. H., and Flagan, R. C.: Design, simulation, and characterization of a radial opposed migration ion and aerosol classifier (ROMIAC), Aerosol Science and Technology, 51, 801–823, https://doi.org/10.1080/02786826.2017.1315046, 2017.

Okuyama, K., Kousaka, Y., and Motouchi, T.: Condensational Growth of Ultrafine Aerosol Particles in a New Particle Size Magnifier, Aerosol Science and Technology, 3, 353–366, https://doi.org/10.1080/02786828408959024, 1984.

Peineke, C., Attoui, M., and Schmidt-Ott, A.: Using a glowing wire generator for production of charged, uniformly-sized nanoparticles at high concentrations, Journal of Aerosol Science, 37, 1651–1661, 2006.

Sgro, L. A. and de la Mora, J. F.: A Simple Turbulent Mixing CNC for Charged Particle Detection Down to 1.2 nm, Aerosol Science and Technology, 38, 1–11, https://doi.org/10.1080/02786820490247560, 2004.

Shah, S. D. and Cocker, D. R.: A Fast Scanning Mobility Particle Spectrometer for Monitoring Transient Particle Size Distributions, Aerosol Science and Technology, 39, 519–526, https://doi.org/10.1080/027868291004652, 2005.

Wang, J., McNeill, V. F., Collins, D. R., and Flagan, R. C.: Fast Mixing Condensation Nucleus Counter: Application to Rapid Scanning Differential Mobility Analyzer Measurements, Aerosol Science and Technology, 36, 678–689, https://doi.org/10.1080/02786820290038366, 2002.

Wang, M., Kong, W., Marten, R., He, X.-C., Chen, D., Pfeifer, J., Heitto, A., Kontkanen, J., Dada, L., Kürten, A., Yli-Juuti, T., Manninen, H. E., Amanatidis, S., Amorim, A., Baalbaki, R., Baccarini, A., Bell, D. M., Bertozzi, B., Bräkling, S., Brilke, S., Murillo, L. C., Chiu, R., Chu, B., De Menezes, L.-P., Duplissy, J., Finkenzeller, H., Carracedo, L. G., Granzin, M., Guida, R., Hansel, A., Hofbauer, V., Krechmer, J., Lehtipalo, K., Lamkaddam, H., Lampimäki, M., Lee, C. P., Makhmutov, V., Marie, G., Mathot, S., Mauldin, R. L., Mentler, B., Müller, T., Onnela, A., Partoll, E., Petäjä, T., Philippov, M., Pospisilova, V., Ranjithkumar, A., Rissanen, M., Rörup, B., Scholz, W., Shen, J., Simon, M., Sipilä, M., Steiner, G., Stolzenburg, D., Tham, Y. J., Tomé, A., Wagner, A. C., Wang, D. S., Wang, Y., Weber, S. K., Winkler, P. M., Wlasits, P. J., Wu, Y., Xiao, M., Ye, Q., Zauner-Wieczorek, M., Zhou, X., Volkamer, R., Riipinen, I., Dommen, J., Curtius, J., Baltensperger, U., Kulmala, M., Worsnop, D. R., Kirkby, J., Seinfeld, J. H., El-Haddad, I., Flagan, R. C., and Donahue, N. M.: Rapid growth of new atmospheric particles by nitric acid and ammonia condensation, Nature, 581, 184–189, https://doi.org/10.1038/s41586-020-2270-4, 2020.

Wiedensohler, A.: An approximation of the bipolar charge distribution for particles in the submicron size range, Journal of Aerosol Science, 19, 387–389, https://doi.org/10.1016/0021-8502(88)90278-9, 1988.